# IDENTICAL INITIALIZATION: A UNIVERSAL APPROACH TO FAST AND STABLE TRAINING OF NEURAL NETWORKS

## ABSTRACT

A well-conditioned initialization is beneficial for training deep neural networks. However, existing initialization approaches do not simultaneously show stability and universality. Specifically, even though the widely-used Xavier and Kaiming initialization approaches can generally fit a variety of networks, they fail to train residual networks without Batch Normalization for calculating an inappropriate scale on data-flow. On the other hand, some literature design stable initialization (e.g., Fixup and ReZero) based on dynamical isometry, an efficient learning mechanism. Nonetheless, these methods are specifically designed for either a non-residual structure or a residual block only, and even include extra auxiliary components, limiting their applicable range. Intriguingly, we find that the identity matrix is a feasible and universal solution to the aforementioned problems, as it adheres to dynamical isometry while remaining applicable to a wide range of models. Motivated by this, we develop Identical Initialization (IDInit), a sufficiently stable, universal, and fast-converging approach to the identity matrix. Empirical results on a variety of benchmarks show that IDInit is universal to various network types, and practically useful with good performance and fast convergence.

## 1 INTRODUCTION

Deep Neural Networks (DNNs) have attracted significant attention due to their versatility in various applications. To obtain well-conditioned training, a suitable initialization is important (Sutskever et al., 2013; Arpit et al., 2019; Huang et al., 2020; Pan et al., 2022). Common initialization methods include Xavier (Glorot & Bengio, 2010) and Kaiming initialization (He et al., 2015). Later, dynamical isometry (Saxe et al., 2014) is widely used for building stable starting status for a very deep network (Mishkin & Matas, 2016; Burkholz & Dubatovka, 2019), and even can stably train 10000-layered networks (Xiao et al., 2018). However, despite the successes, these aforementioned methods are unsuitable for residual blocks without Batch Normalization since calculating an improper scale on data-flow (see Sec. 4.2 and Sec. C.2). Addressing this issue, Bachlechner et al. (2021); Hardt & Ma (2017) proposed to stabilize the training of residual blocks according to dynamical isometry mechanism by transiting identity via multiplying residual stem with 0. Nevertheless, the identity-maintaining methods can be only applied on residual networks, and some of them even require auxiliary components to stabilize model training (Blumenfeld et al., 2020; Bachlechner et al., 2021; Zhao et al., 2021), leading to lost generality to other network structures. To overcome the above problems, in this paper, we propose a stable, general, and fast-converging initialization approach, based on the identity matrix.

**Motivation on Identity Matrix.** As mentioned above, implementing identity transition can naturally correspond to the isometric mechanism that is beneficial for fast convergence and improving performance (Bachlechner et al., 2021). To maintain this transition of both residual and non-residual modules, the identity matrix is a potential solution. In detail, consider $i$-th block in a DNN,

$$x^{(i+1)} = (r + \prod_{j=1}^{m} \theta^{(i,j)}) x^{(i)}, \tag{1}$$

where $m$ is the number of weights in a block, $\theta^{(i,j)}$ means $j$-th weight in the $i$-th block, and $r \in \{0, I\}$ indicates whether Eq. (1) denotes a residual layer. Usually when $r = 0$ and $m = 1$, Eq. (1) is

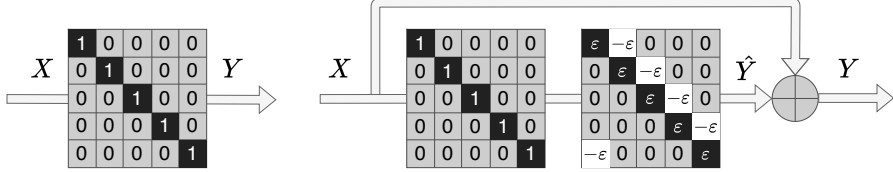

Figure 1: A simple case of IDInit on a residual and non-residual network layer. $\varepsilon$ is set to $1e - 6$, therefore, $\hat{Y} \approx 0$. $Y$ of both left and right sub-figures is equal to $X$.

a non-residual layer. Under this condition, to achieve the identity transition, namely $x^{(i+1)} = x^{(i)}$, $\theta^{(i,1)} = I$ is the unique solution. When $r = I$ and $m \geq 2$, Eq. (1) denotes a residual layer. At this moment, setting all $\{\theta^{(i,j)}\}_{j=1}^{m}$ to $I$ is not feasible. Therefore, it is usual to set the last weight $\theta^{(i,m)}$ to $0$ to maintain identity (Zhang et al., 2019; Zhao et al., 2021). Overall, without loss of generality, it is necessary and feasible to apply an identity matrix as an initial status for maintaining identity.

**Identical Initialization (IDInit).** In this paper, we introduce the identity matrix into designing a novel and practical initialization, named Identical Initialization (IDInit). A simple case of IDInit is shown in Figure 1. For a non-residual condition, squared identity matrices can easily achieve identity transition, however, the most general condition, non-squared matrices cannot (Vaswani et al., 2017). To adapt the non-squared situation, we modify the identity matrix by maintaining signal variance as described in Sec. 3.2. For a residual condition, a dead neuron problem will be caused by directly setting the last weight in a residual stem to 0 (Zhang et al., 2019; Zhao et al., 2021). Addressing this, we simply select some elements to an extremely small numerical value $\varepsilon$ to increase trainable neurons as in Figure 1. We also observe that an identical convolution layer that maintains identity transit degrades performance dramatically. Tackling the issue, we propose a modest change to an identical convolution layer by fusing spatial information, leading to significant improvement. Moreover, we find the model performance is uncertain with random initialization whose values are sampled from a probability distribution (Glorot & Bengio, 2010; He et al., 2015). IDInit is exactly able to handle this situation of its determinacy that is unique, and uncorrelated with samplers.

To our best knowledge, it is the first trial to put identical-like initialization into practice. Previous work, Bartlett et al. (2019), has used the identity matrix as a weight with excessive attention on approximate analysis, resulting in imprecise conclusions for the gap between reality and theory. Additionally, this work addresses only relatively simple scenarios without an activation function and ignores momentum in stochastic gradient descent (SGD), which can optimize critical points to a degree. And this method is incompatible with ResNets without Batch Normalization (Ioffe & Szegedy, 2015), since signals will explode if all weights are set identically. Motivated by some of the observations and addressing shortcomings, we provide a simple, practical, and efficient initialization method.

## 2 BACKGROUND AND RELATED WORK

**Dynamical Isometry.** Give an $L$-layer network with blocks formulated by Eq. (1), $x^{(0)}$ is input and $x^{(L)}$ is the $L$-th layer's output. Assuming signal magnitude (e.g., $\sigma^2(x^{(i)})$) of each layer changing in the scale $\alpha$, the last signal magnitude can reach $\alpha^L$ (e.g., $\sigma^2(x^{(L)}) = \alpha^L \sigma^2(x^{(0)})$), making it easy to cause signal explosion and diffusion, especially for large $L$. Introduced from mean-field theory (Pennington et al., 2017; 2018), the dynamical isometry is a comparably reasonable mechanism to measure models' trainability. To utilize this paradigm, it usually considers the input-output Jacobian

$$J_{io} = \frac{\partial x^{(L)}}{\partial x^{(0)}}, \tag{2}$$

whose mean squared singular value is $\chi$. (Pennington et al., 2017) and (Bachlechner et al., 2021) show that $\chi > 1$ indicates the model in a chaotic phase, and back-propagated gradients will explode exponentially. By contrast, $\chi < 1$ means a model in an ordered manner that back-propagated gradients exponentially vanish. $\chi = 1$ is a critical line of initialization, avoiding vanishing or

exploding gradients. Using the stable isometry condition can provide a sufficiently stable training process for networks (Gilboa et al., 2019; Poole et al., 2016; Yang & Schoenholz, 2017).

**Related Initialization.** Xavier (Glorot & Bengio, 2010) and Kaiming initialization (He et al., 2015) is classical initialization. Specially for residual network efficiency, Hardt & Ma (2017) theoretically demonstrates that network training benefits from keeping identity. Then, maintaining identity transit turns into a promising way to further improve model performance. Fixup (Zhang et al., 2019) and ZerO (Zhao et al., 2021) both set residual stem to 0 (not residual connections) to guarantee the identity of signals, thereby initializing ResNets successfully. SkipInit (De & Smith, 2020) replaces Batch Normalization with a multiplier whose value is 0. ReZero (Bachlechner et al., 2021) directly adds extra parameters of value 0 to keep identity, leading to fast convergence.

## 3 IDENTICAL INITIALIZATION (IDINIT)

In this section, we first use a descriptive example to elaborate on the mechanism of IDInit for fast convergence. Then, we describe the way to maintain the stability for non-square identity-like matrices. Next, we propose a zero preserving scheme to solve the dead neuron problem in identity transition. At last, we use a simple but efficient reshaping strategy to significantly enhance convolution performance.

### 3.1 TOY EXAMPLE

Following Bachlechner et al. (2021), we utilize a simple example of the mechanism that dynamical isometry helps IDInit to obtain a fast convergence. Considering a $L$-layer network with simplified Eq. (1):

$$x^{(L)} = (r + w^{(2)}w^{(1)})^L x^{(0)}, \quad (3)$$

where $w^{(1)}$ and $w^{(2)}$ denote the first weight and last weight in a residual stem respectively, and $x^{(*)}$ is the feature in layers. $r \in \{0, 1\}$ determines residual connection. Specifically, $r = 0$ and $r = 1$ represent non-residual and residual conditions respectively. The Jacobian of Eq. (3) is $J_{0L} = (r + w^{(2)}w^{(1)})^L$.

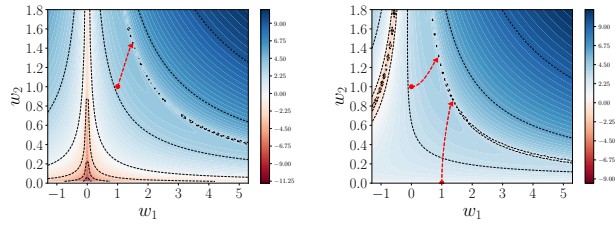

(a) Non-Residual Plot.      (b) Residual Plot.

Figure 2: Contour plots of the log gradient norm $\log ||\partial R||_2$ on non-residual and residual networks. $w^{(1)}$ and $w^{(2)}$ are both weights. The training process set as Bachlechner et al. (2021), which is conducted on ground-truth $x^{(L)} = 50 \times x_0$ via gradient descent using a training set of $x_0 = \{1., 1.1, ..., 1.8\}$. (a) shows $\{w^{(2)} = w^{(1)} = 1\}$ can avoid poorly conditioned regions around 0, and converge to $w^{(1)}w^{(2)} = 2.19$. (b) cares about two initial position $\{w^{(1)} = 0, w^{(2)} = 1\}$ and $\{w^{(2)} = 1, w^{(1)} = 0\}$. The two points' trajectories do not also pass the poor regions around $w^{(1)} = -1, w^{(2)} = 1$ and converge to the solution $w^{(1)}w^{(2)} = 1.19$.

Obviously, identity transition on both non-residual and residual settings, namely $\{r = 0, w^{(2)} = w^{(1)} = 1\}$ and $\{r = 1, w^{(1)} = 1, w^{(2)} = 0\}$ respectively, will achieve $J_{0L} = 1$, which conforms to the dynamical isometry mechanism that helps improving training ability (Pennington et al., 2017). Further, we delve into a gradient update analysis. Following gradient descent, $w_1$ can be updated by

$$w^{(1)} = w^{(1)} - \lambda L w^{(2)} x^{(0)} (r + w^{(2)}w^{(1)})^{L-1} \partial_x R(x)|_{x=x^{(L)}}, \quad (4)$$

where $R$ means the loss function, and $\lambda$ is a learning rate. As $w^{(1)}$ and $w^{(2)}$ are equivalent in Eq. (3), $w^{(2)}$ can be updated similar to Eq. (4). When $w^{(1)} = 1$, updates are required less than 1. Therefore, the learning rate is constrained to

$$\begin{cases} \lambda \propto L^{-1}, & \text{if non-residual,} \\ \lambda \propto L^{-1}(1 + w^{(2)})^{L-1}, & \text{if residual.} \end{cases} \quad (5)$$

For the non-residual condition, the learning rate is polynomial to $L$, thereby insensitive to the depth. By contrast, in the residual block, $w^{(2)} >> 0$ will cause learning rate exponentially small and $w^{(2)} = -1$ also cause gradient diffusion. On this condition, setting $w^{(2)} = 0$ can be a good solution

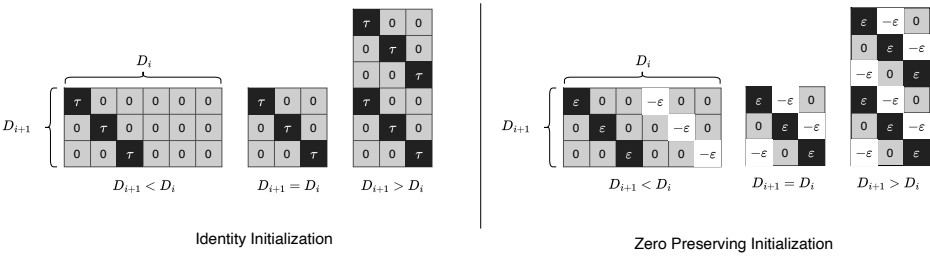

Figure 3: Illustration of IDInit for fully-connected layers.

for avoiding large output and restricting gradients in a suitable norm. Besides, it is feasible to update $w^{(2)}$ with the first non-trial step

$$w^{(2)} = -\lambda L w^{(1)} x^{(0)} \partial_x R(x)|_{x=x^{(L)}}, \tag{6}$$

and will converge with a learning rate that is polynomial in the depth $L$ of the network. We plot the training dynamics in Figure 2, and use this simple example to illustrate the mechanism of IDInit, which is always a well-conditioned position for training.

### 3.2 IDENTICAL FULLY-CONNECTED LAYERS

A standard identity matrix can naturally satisfy identity transition as illustrated in Figure 1. However, in a non-square situation, this natural advantage is lost. To address this problem, we give a small modification to fit a non-square matrix. Specifically, for a fully-connected layer transformed from Eq. (1) as $x^{(i+1)} = \theta^{(i)} x^{(i)}$, we set the weight $\theta^{(i)} \in \mathbb{R}^{D_{i+1} \times D_i}$ to

$$\theta^{(i)}_{m,j} = \begin{cases} \tau, & \text{if } m \equiv j \pmod{D_i}, \\ 0, & \text{otherwise.} \end{cases} \tag{7}$$

The initialization formulated as Eq. (7) is termed as $\text{IDI}_\tau$, where IDI means the identical initialization function, and $\tau$ is calculated by considering the activation function, e.g., $\tau_{ReLU} = \frac{1}{2}$ and $\tau_{sigmoid} = 1$ for ReLU and sigmoid activation functions respectively. As in Figure 3, setting $\tau = 1$ can form $\text{IDI}_1$ initialization. Moreover, We provide a trainability analysis on the non-squared condition of $\text{IDI}_\tau$ in Sec. A.2 of the appendix.

**Remark.** ZerO constructs a dimension-increasing matrix (i.e., $D_{i+1} > D_i$) by padding zero values to an identity matrix, which is similar to our method, namely padding identity matrices recurrently. While both the two padding strategies can transit identity propagation, padding zero values will cause zero gradients, which is known as a rank constraint problem, and can severely affect the performance. By contrast, $\text{IDI}_\tau$ can derive non-zero gradients, thereby, inherently avoiding the rank constraint problem. Therefore, $\text{IDI}_\tau$ has significant improvement on the prior padding method.

### 3.3 IDENTICAL RESIDUAL BLOCKS

At present, residual blocks become the most popular module in almost all the neural network (e.g., Mixers (Liu et al., 2021a; Tolstikhin et al., 2021), Convolutions (He et al., 2016; Zhang et al., 2019), and Transformers (Liu et al., 2021b; Vaswani et al., 2017)). A residual block is usually constructed with a residual connection and several transformations in the residual stem. Following Eq. (1), a simple residual case with two parameter matrices can be formulated as

$$x^{(i+1)} = (I + \theta^{(i,0)} \theta^{(i,1)}) x^{(i)}, \tag{8}$$

where $x^{(i)}$ means an input of $i$-th residual block in a network, $I$ is an identity matrix denoting residual connection, and $\theta^{(i,0)}$ and $\theta^{(i,1)}$ are weights in the $i$-th residual stem of a residual block.

Recent research (Zhang et al., 2019; Zhao et al., 2021) directly sets (i) the last transformation in the residual stem to 0, i.e., $\theta^{(i,0)} = 0$, thereby maintaining an identity as

$$x^{(i+1)} = (I + 0) x^{(i)} = x^{(i)}; \tag{9}$$

(ii) the last classification layer of the network to 0. Although the two settings work, no interpretation is provided for the trainability. We consider the effect of (i) is from dynamical isometry, since the mean squared singular value $\chi$ of the input-output Jacobian of the network is 1. As for the setting (ii), the last layer can be trained with arbitrary values when transiting identity in networks, according to the trainability analysis in Sec. 3.2. Therefore, when setting all values of the last classification layer to 0, this classification layer can still obtain non-zero gradients of a suitable magnitude. Since the settings (i) keeps identity transition and (ii) helps to improve training stability, we also consider adopting them to our initialization. However, while there is no problem with the setting (ii), the setting (i) contradicts the proposed identical initialization as generating a dead neuron problem.

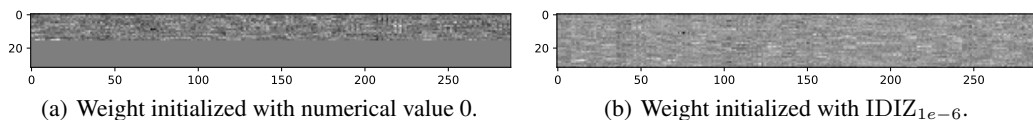

(a) Weight initialized with numerical value 0.       (b) Weight initialized with $\text{IDIZ}_{1e-6}$.

Figure 4: The last weight in a residual block of a trained ResNet. More than half of elements in (a) are not trained, which is known as the dead neuron. By contrast, $\text{IDIZ}_{1e-6}$ successfully solves the dead neuron problem and make all the elements in (b) trainable.

**Dead Neuron Problem.** Fixup (Zhang et al., 2019) only uses a multiplier of 1 after $\theta^{(i,0)} = 0$, thereby obtaining non-zero gradients. However, in a realistic implementation of neural networks, the the multiplier of Batch Normalization can be set to 0 (Goyal et al., 2017), and down-sampling operation can also cause 0 filled features[2]. Under the implementations, $\theta^{(i,0)}$ always acquires gradients with 0 values, known as the dead neuron problem, causing failed weight updating.

To overcome the problem, we generate small values on $\theta^{(i,0)}$ to assist in training. Recall the goal of setting (i) that outputs 0. Therefore, we consider building a calculation to get the expectation and variance of outputs approaching to 0. Prior to that, we consider two i.i.d variables, $v_1$ and $v_2$, whose variances are $\sigma^2(v_1) = \sigma^2(v_2) = \varphi$ and means are $\mu(v_1) = \mu(v_2) = \gamma$. The variable $v = \varepsilon(v1 - v2)$ have

$$\begin{cases} \mu(v) = 0, \\ \sigma^2(v) = 2\varphi\varepsilon^2, \end{cases} \quad (10)$$

where $\varepsilon$ is a coefficient, and $\sigma^2(v)$ will be limited to 0 when $\varepsilon$ is sufficiently small. Assuming elements of $x^{(i)}$ are i.i.d to each other, by applying subtraction on any two elements, the result has a mean of 0, and a variance related to $\varepsilon$. We also take $\theta^{(i,0)} \in \mathbb{R}^{D_{i+1} \times D_i}$ as an instance. At first, we initialize $\theta^{(i,0)}$ with $\text{IDI}_\varepsilon$. Then consider two cases: (i) if $D_{i+1} < D_i$, setting $\theta^{(i)}_{:,D_{i+1}+1:D_i}$ with $\text{IDI}_{-\varepsilon}$; (ii) if $D_{i+1} \geq D_i$, set $\theta^{(i)}_{m,j} = -\varepsilon$, when $m\%D_i = j - 1$. Therefore, we can obtain a variance of 0 by setting $\varepsilon$ to a small value. This method is termed as $\text{IDIZ}_\varepsilon$. In this paper, we set $\varepsilon = 1e - 6$ everywhere. As shown in Figure 4, $\text{IDIZ}_{1e-6}$ successfully initializes the last weight in a residual block.

### 3.4 IDENTICAL CONVOLUTIONAL LAYERS

Besides the fully-connected layer, the convolutional layer is another important structure in deep neural networks. For constructing a general initialization, we also explore an initialization pattern for convolution with identity transition. In this part, we consider a convolution layer formulated as

$$\mathcal{Y} = \mathcal{C} \otimes \mathcal{X} + \mathcal{B}, \quad (11)$$

where $\mathcal{X} \in \mathbb{R}^{h \times w \times c_{in}}$ is an input tensor, $\mathcal{Y} \in \mathbb{R}^{h' \times w' \times c_{out}}$ is an output tensor, $\mathcal{C} \in \mathbb{R}^{k \times k \times c_{in} \times c_{out}}$ is a convolutional kernel, and $\mathcal{B} \in \mathbb{R}^{h' \times w' \times c_{out}}$ is a bias and usually set to 0. $h, w$ and $h', w'$ denote height and width of the input and the output, respectively. $c_{in}$ and $c_{out}$ are channels of input and output separately. $k$ means convolutional kernel size.

---

[1]https://github.com/hongyi-zhang/Fixup/blob/master/cifar/models/resnet_cifar.py
[2]https://github.com/akamaster/pytorch_resnet_cifar10/edit/master/resnet.py

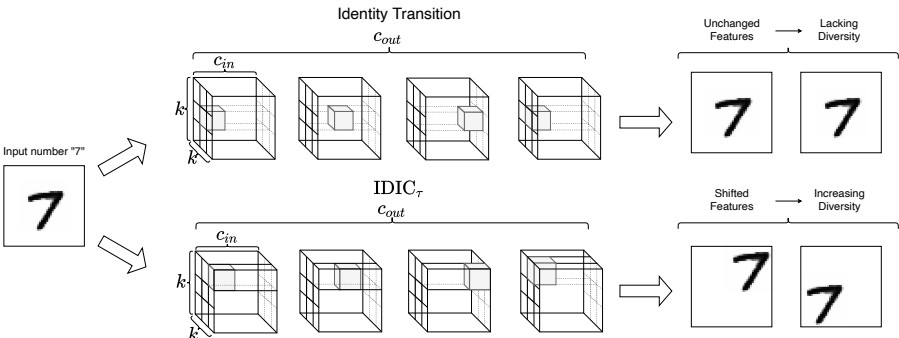

Figure 5: A case of number "7" on Identical Convolution Layer. The upper sub-figure maintains the identity transition. The under sub-figure is $\text{IDIC}_\tau$ initialization that shifts features for increasing diversity. More feature diversity from $\text{IDIC}_\tau$ is beneficial for improving model performance.

**Degeneration from Convolution Identity Transition.** As shown upper sub-figure in Figure 5, similar to an identity matrix, a convolution layer can transit identity by setting 0-filled $\mathcal{C}$ as

$$\text{IDI}_1(\mathcal{C}_{n,n,:,:}), \tag{12}$$

where $n \in \mathbb{N}^+$ and $k = 2n+1$. As a convolutional kernel window size, $k$ is usually an odd number. After Eq. (12), $\mathcal{Y} = \mathcal{X}$ if $c_{in} = c_{out}$. $\mathcal{Y}$ will under-sample and over-sample on $\mathcal{X}$ along channel when $c_{in} > c_{out}$ and $c_{in} < c_{out}$ respectively. Keeping identity is usually considered an efficient way to improve model performance, however, we find that this setting can lead to a fatal performance degeneration (see Sec. 4.3).

Inspired by Han et al. (2020) that enhance model performance by increasing feature diversity, we propose to fuse spatial information by simply reshaping a matrix initialized $\text{IDI}_\tau$. Specifically, consider one inner product part of a convolution on feature patch $Y^{(patch)} \in \mathbb{R}^{c_{out}}$ as

$$Y^{(patch)} = CX^{(patch)}, \tag{13}$$

where $C \in \mathbb{R}^{c_{out} \times kkc_{in}}$ is the convolutional kernel, and $X^{(patch)} \in \mathbb{R}^{kkc_{in}}$ is input patch. By initializing $C$ with $\text{IDI}_\tau$, then reshaping $C$ into $\mathcal{C} \in \mathbb{R}^{k \times k \times c_{in} \times c_{out}}$, our initialization for a convolution is completed. As shown in the under sub-figure of Figure 5, spatial features are shifted, thereby increasing feature diversity. We utilize $\text{IDIC}_\tau$ to denote such a reshaping process. In addition, we use $\text{IDIZC}_\varepsilon$ to represent the similar operation on $\text{IDIZ}_\varepsilon$.

We conclude the whole IDInit as (1) **Non-Residual Networks.** Directly applying $\text{IDI}_\tau$ and $\text{IDIC}_\tau$ to all the fully-connected and convolutional layers, respectively; (2) **Residual Networks.** (i) Applying $\text{IDI}_\tau$ and $\text{IDIC}_\tau$ to all the fully-connected and convolutional layers, respectively; (ii) Applying $\text{IDIZ}_\varepsilon$ and $\text{IDIZC}_\varepsilon$ to the fully-connected and convolutional layers in the last position of residual blocks, and the position of last classification layer.

## 4 EXPERIMENT

In this section, we design a set of experiments to validate the proposed IDInit. To begin with, we conduct experiments on non-residual convolution and residual convolution in Sec. 4.1 and Sec. 4.2 respectively. Then we implement an ablation experiment in Sec. 4.3 to show the effect of the proposed two modifications in Sec. 3. Later we analyze the determinacy of initialization in Sec. 4.4. At last, we employ experiments on tasks large-scale datasets ImageNet and WuDaoCorpora in Sec. 4.5 and Sec. 4.6 separately. We also analyze the variance amplification in Sec. C.2 in Appendix.

### 4.1 VALIDATION ON NON-RESIDUAL CONVOLUTION

We use this experiment to show the hardness of classical Kaiming initialization for training. This experiment is conducted on Cifar10. We use nine convolutional layers named AllConv (Springenberg et al., 2015). We show the structure of AllConv in Table 4 in the appendix. The optimizer is

Stochastic Gradient Descent (SGD) with momentum 0.9, weight decay 5e-4, and learning rate 1e-1. The learning rate scheduler adopts a warm-up cosine reduction strategy. We run the model in 300 epochs on one Nvidia A100. We adopt Kaiming initialization and IDInit w/o $IDIC_\tau$ initialization for comparison.

Results are shown in Figure 6, without a warm-up strategy which is a strong trick for training, both Kaiming and IDInit w/o $IDIC_\varepsilon$ fail to train the model. By contrast, our initialization can train AllConv and maintain the highest performance in all situations, showing a strong effect on stability and performance. As IDInit w/o $IDIC_\varepsilon$ performs awfully, we demonstrate the degeneration from identity transition mentioned in Sec. 3.4 is relieved by our new reshaping strategy.

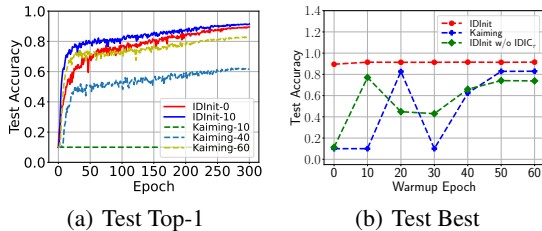

(a) Test Top-1    (b) Test Best

Figure 6: Results of AllConv on Cifar10. The number behind the initialization denotes the warm-up epochs.

## 4.2 VALIDATION ON RESIDUAL CONVOLUTION

In this experiment, we validate the proposed initialization with the comparison with existing initialization, including (1) Fixup; (2) SkipInit; (3) ReZero; (4) Kaiming; (5) Zero $\gamma$ (Setting the scale in Batch Normalization (BN) to 0) (Goyal et al., 2017). We use ResNet-56/110 as backbones on Cifar10. To verify the universality, we use two settings, namely w/ and w/o BN. For analyzing convergence, we adopt both SGD and Adam optimizer for updating models. We set SGD, with the momentum 0.9, the weight decay 5e-4, and the learning rate 0.2. For Adam, the learning rate is 0.001, $\beta_1$ is 0.9 and $\beta_2$ is 0.999. We train models for 200 epochs. The learning rate is reduced with a cosine function.

Table 1: Results on Cifar10. A multiplier is used for scaling the output of residual stem. For validating universality, we consider "ResNet w/o Multiplier" which is a tough condition for training.

| | | 56 Layer (SGD/Adam) | | 110 Layer (SGD/Adam) | |
|---|---|---|---|---|---|
| Model | Init. | Acc. | Epochs to 80% Acc. | Acc. | Epochs to 80% Acc. |
| ResNet w/ Multiplier | Zero $\gamma$ | $92.32_{\pm0.19}/87.37_{\pm0.43}$ | $57_{\pm7}/63_{\pm4}$ | $93.07_{\pm0.28}/88.30_{\pm0.31}$ | $36_{\pm2}/56_{\pm7}$ |
| | Fixup | $93.24_{\pm0.82}/\underline{89.50}_{\pm0.18}$ | $\underline{31}_{\pm3}/55_{\pm3}$ | $93.32_{\pm0.23}/\mathbf{90.67}_{\pm0.12}$ | $33_{\pm3}/49_{\pm2}$ |
| | SkipInit | $92.29_{\pm0.30}/85.45_{\pm0.74}$ | $\mathbf{26}_{\pm1}/81_{\pm3}$ | $92.67_{\pm0.16}/87.18_{\pm0.94}$ | $\underline{31}_{\pm5}/70_{\pm7}$ |
| | ReZero | $93.06_{\pm0.54}/89.26_{\pm0.30}$ | $33_{\pm2}/\underline{44}_{\pm3}$ | $94.03_{\pm0.26}/90.25_{\pm0.20}$ | $35_{\pm5}/\underline{38}_{\pm3}$ |
| | Kaiming | $\underline{93.36}_{\pm0.14}/87.55_{\pm0.32}$ | $34_{\pm3}/50_{\pm2}$ | $\mathbf{94.06}_{\pm0.18}/87.89_{\pm0.41}$ | $33_{\pm4}/56_{\pm3}$ |
| | IDInit | $\mathbf{93.41}_{\pm0.10}/\mathbf{90.01}_{\pm0.32}$ | $\mathbf{26}_{\pm1}/\mathbf{34}_{\pm1}$ | $\underline{94.04}_{\pm0.24}/\underline{90.53}_{\pm0.10}$ | $\mathbf{27}_{\pm1}/\mathbf{36}_{\pm2}$ |
| ResNet w/o Multiplier | Kaiming | Diverged/$63.84_{\pm0.90}$ | - / - | Diverged/$16.45_{\pm1.45}$ | - / - |
| | IDInit | $91.53_{\pm0.21}/86.29_{\pm0.40}$ | $61_{\pm1}/58_{\pm2}$ | $91.51_{\pm0.14}/87.20_{\pm0.30}$ | $57_{\pm2}/49_{\pm1}$ |

Results are shown in Table 1. The proposed IDInit can achieve the least epochs to reach 80% accuracy in all settings, which shows a good convergence ability. In the w/ BN setting, IDInit consistently performs well compared with baselines. In the w/o BN setting, Kaiming fails to train the models with SGD optimizer for calculating improper scales for data-flow. Even though using the adaptive Adam optimizer, Kaiming still cannot derive normal results. By contrast, IDInit can work normally in both optimizers and derive comparably reasonable results.

## 4.3 ABLATION EXPERIMENT

We conduct this experiment to validate the effect of the proposed two improvements. The dataset is Cifar10 and the backbone is ResNet-20. We run four times following settings: (i) IDInit w/o $IDIC_\tau$ and w/o $IDIZC_\varepsilon$; (ii) IDInit w/o $IDIC_\tau$ and w/ $IDIZC_\varepsilon$; (iii) IDInit w/ $IDIC_\tau$ and w/o $IDIZC_\varepsilon$; (iv) IDInit. We choose SGD with momentum 0.9, weight decay 5e-4 and learning rate 0.1 to train

the models for 200 epochs. The learning rate is reduced with a cosine function. And data-augment mixup is applied.

As shown in Table 2, by applying the identity matrix directly, (i) obtains the lowest accuracy of 87.01% among all cases. Regarding results of (ii) and (iii), both the two settings can make significant improvements of nearly 5.89% and 3.42% from (i), respectively. And $IDIZC_\varepsilon$ can make a deeper effect than $IDIC_\tau$. Equipping $IDIC_\tau$ and $IDIZC_\varepsilon$, IDInit will improve performance further, which demonstrates our modification is efficient and practical.

Table 2: Results of the ablation experiment on ResNet-20.

| Setting | (i) | (ii) | (iii) | (iv) |
|---|---|---|---|---|
| Accracy | $87.01 \pm 0.29$ | $92.9 \pm 0.18$ | $90.43 \pm 0.14$ | $93.22 \pm 0.05$ |

## 4.4 DETERMINACY ANALYSIS ON TEXT CLASSIFICATION

We conduct this experiment to show random initialization can cause unacceptable uncertainty measured with standard derivation, and IDInit can solve this problem for the inherent determinacy that seeds will not change initial weights when a model is determined. We implement text classification on SST2 and SST5 (Socher et al., 2013) and select TextCNN (Kim, 2014), TextRNN (Lai et al., 2015) and Transformer (Vaswani et al., 2017) for comparison. For TextCNN and TextRNN, we use AdaDelta (Zeiler, 2012) optimizer with a learning rate 1.0 and adopt Adam (Kingma & Ba, 2015) for Transformer with a learning rate 1e-4. For the embedding layer, we utilize Glove (Pennington et al., 2014) and Word2Vec(Mikolov et al., 2013) to initialize the embedding weights. All models are trained up to 10 epochs. Fixing other hyper-parameters except for the initialization, we run all the random initialization for 5 times.

Table 3: Results of TextCNN and TextRNN on SST2 and SST5. The subscript G denotes the embedding layer is initialized by Glove while W indicates Word2Vec. "Default" means the default initialization of models, specifically, Kaiming for TextCNN, and Xavier for both TextRNN and Transformer. Std values larger than 1.0 are marked in red.

| Datasets | Init. | $TextCNN_{G/W}$ | $TextRNN_{G/W}$ | $Transformer_{G/W}$ |
|---|---|---|---|---|
| SST2 | Default | $81.40_{\pm0.66}/84.56_{\pm0.43}$ | $81.69_{\pm0.30}/84.29_{\pm0.70}$ | $80.97_{\pm1.20}/83.36_{\pm0.76}$ |
| | Orthogonal | $82.24_{\pm0.44}/84.37_{\pm0.38}$ | $81.86_{\pm0.55}/84.61_{\pm0.78}$ | $82.22_{\pm0.87}/83.99_{\pm0.23}$ |
| | IDInit | **83.14/86.05** | **83.14/86.22** | **82.66/85.13** |
| SST5 | Default | $44.68_{\pm0.88}/46.15_{\pm0.62}$ | $44.27_{\pm0.88}/47.04_{\pm0.48}$ | $41.81_{\pm1.17}/44.02_{\pm1.27}$ |
| | Orthogonal | $44.91_{\pm0.81}/46.76_{\pm0.68}$ | $44.61_{\pm1.18}/46.13_{\pm0.79}$ | $43.01_{\pm1.61}/44.92_{\pm1.52}$ |
| | IDInit | **45.66/48.05** | **47.12/48.83** | **45.22/46.24** |

As shown in Table 3, all the initialization methods can work normally. However, default random initialization obtains the lowest accuracy in both SST2 and SST5. Orthogonal initialization only performs slighter better than IDInit with TextRNN on SST2, and may cause a bad result as 42.35% with TextCNN on SST5. On the contrary, the proposed IDInit can achieve the highest accuracy in almost all the conditions, which shows the sufficient efficiency of IDInit. Furthermore, as a deterministic method, IDInit offers the ability to develop a fully deterministic training framework as well as improve performance for easy-to-analyze structures.

## 4.5 IMAGE CLASSIFICATION ON IMAGENET

In this experiment, we use ViT-B/32 (Dosovitskiy et al., 2021), ResNet-50/152 (RN-50/152) and Se-ResNet-50 (SRN-50) as backbones on ImageNet. For ViT-B/32, the optimizer is AdamW with a learning rate 1e-3 and a weight decay 5e-2. The training epochs is 300. We use 30 epochs for warm-up. For RN-50/152 and SRN-50, the optimizer is SGD with a learning rate 1e-1 and a weight decay

1e-4. The epoch for training is 90. We use 9 epochs for warm-up. For all models, the batch size is 1024, and we apply data-augment including cutmix (Yun et al., 2019) with $\alpha = 1.0$, mixup (Zhang et al., 2018) with $\alpha = 0.8$, the switching probability is 0.5 and a label smoothing with 0.1.

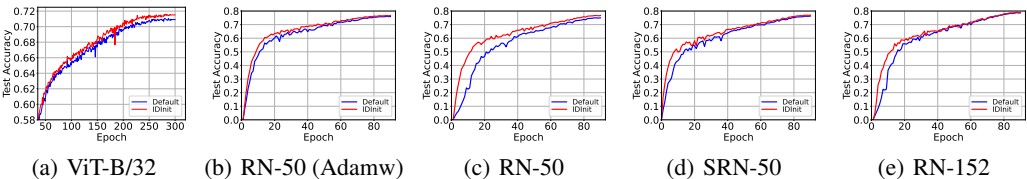

(a) ViT-B/32  (b) RN-50 (Adamw)  (c) RN-50  (d) SRN-50  (e) RN-152

Figure 7: Results on ImageNet. "Default" means the default initialization of models. RN-50 (Adamw) means that ResNet-50 is trained with the same optimizer Adamw as the ViT-B/32.

Results are shown in Figure 7. On three types of networks, i.e., ViT, ResNet and Se-ResNet, and multiple depths, IDInit always achieves faster convergence and better performance than the baseline. And when training RN-50 with Adamw, the convergence of IDInit is still consistently fast. This experiment shows the good practicability and promising probability of IDInit, which is beneficial to the artificial intelligence community.

## 4.6 PRE-TRAINING ON WUDAOCORPORA

Pre-training plays an important role in various applications. In this experiment, we pre-train GPT with 8 and 96 layers from scratch on the large-scale text dataset WuDaoCorpora (Yuan et al., 2021) which has 200 GB training data and 72 billion Chinese characters. The optimizer is Adam with a learning rate 1e-3 and a weight decay 5e-4. The batch size is 1536. For 8-layer GPT, we run 70k steps. For 96-layer GPT, we run 13k steps. This experiment is conducted on 32 NVIDIA V100s.

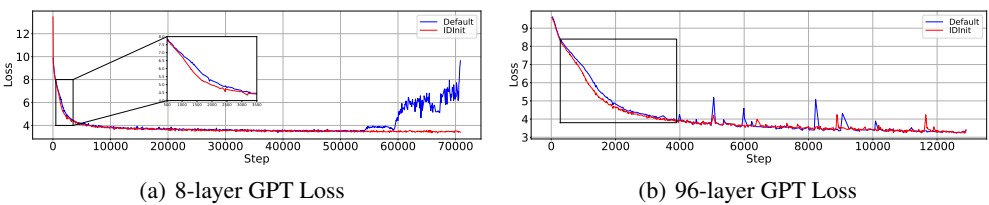

(a) 8-layer GPT Loss  (b) 96-layer GPT Loss

Figure 8: Results of GPT on WuDaoCorpora. "Default" means the default initialization of models for GPT. IDInit is our identical initialization.

As shown in Figure 8, IDInit achieves faster convergence on both 8 and 96 depths. Additionally, as in Figure 8(a), the random initialization's loss unexpectedly increases sharply at around 54k step, indicating an unstable training. By contrast, IDInit can steadly be trained normally. This experiment shows good convergence and stability of IDInit, which can benefit pre-training to some extent.

## 5 CONCLUSION

In this paper, we introduce an identical initialization (IDInit) that is based on the identity matrix. Addressing the problems encountered when developing IDInit, i.e., dead neurons and performance degeneration, we give two concise solutions, namely using zero to wipe off dead neurons and reshaping an identity-like matrix into a tensor thus increasing feature diversity, leading to a performance improvement. With good performance on wide generality, high stability, and fast convergence, IDInit is promising to be applicable in practice. In the future, we hope that this identical design can motivate the AI community to implement more novel initialization methods.

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

# Appendix

## Table of Contents

## A  IDInit Details

### A.1  Full IDInit Scheme

Here, we show the full IDInit scheme in Figure 9.

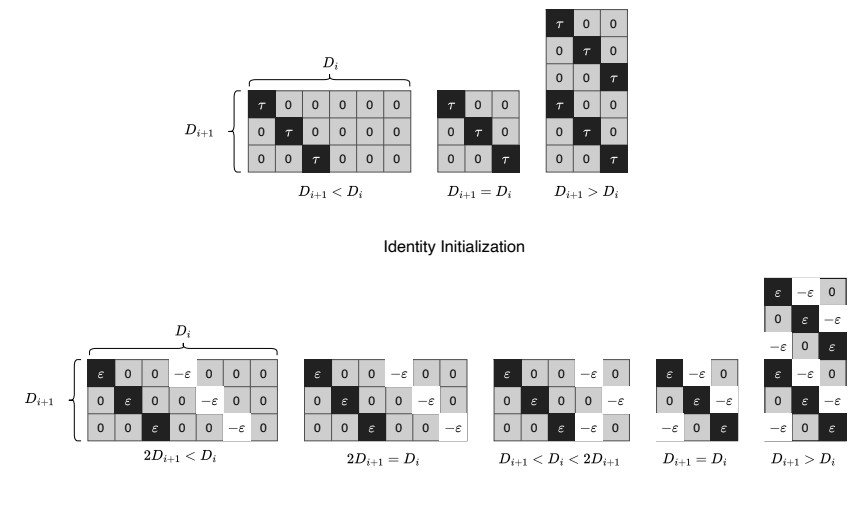

Figure 9: Illustration of IDInit with all conditions.

## A.2 TRAINABILITY ANALYSIS OF NON-SQUARED MATRICES.

The magnitude of signals transiting in layers is usually related to stability, and the variance of signals can be a good indicator of the magnitude according to the variance-control mechanism. Obviously, $\text{IDI}_\tau$ can maintain forward signal variance. Therefore, we explore the trainability from the angle of the gradient backward procedure. Prior to that, consider a network $x^{(L)} = f(x|\Theta) = \theta^{(L-1)}\theta^{(L-2)}\ldots\theta^{(0)}x^{(0)}$, the gradient relationship can be formulated as

$$\frac{\partial \mathcal{L}}{\partial x^{(i)}} = \theta^{(i)\text{T}}\frac{\partial \mathcal{L}}{\partial x^{(i+1)}} = \theta^{(i)\text{T}}\theta^{(i+1)\text{T}}\ldots\theta^{(L-1)\text{T}}\frac{\partial \mathcal{L}}{\partial x^{(L)}}, \qquad \frac{\partial \mathcal{L}}{\partial \theta^{(i)}} = \frac{\partial \mathcal{L}}{\partial x^{(i+1)}}x^{(i)\text{T}}, \qquad (14)$$

where $x^{(i)} \in \mathbb{R}^{D_i}$, $\theta^{(i)} \in \mathbb{R}^{D_{i+1}\times D_i}$ and $\mathcal{L}$ denotes a loss of training. Since $\forall i, D_{i+1} = D_i$, the training process will be stable as that gradients transit identically. Here, we pay attention to non-square conditions. $\tau = 1$ for there is no activation function.

If $\forall i, D_{i+1} < D_i$, then $\frac{\partial \mathcal{L}}{\partial x^{(i)}}$ ($i \in \{1, 2, \ldots, L-1\}$) contains elements of 0, which indicates that $\frac{\partial \mathcal{L}}{\partial \theta^{(i)}}$ must contain 0 values, causing failed updating. However, this is a fake dead phenomenon, since all weights will be trained after several updating steps. At the first training step, $\frac{\partial \mathcal{L}}{\partial \theta^{(L-1)}} = \frac{\partial \mathcal{L}}{\partial x^{(L)}}x^{(L-1)\text{T}}$. For both $\frac{\partial \mathcal{L}}{\partial x^{(L)}}$ and $x^{(L-1)}$ not definitely containing 0 values, $\theta^{(L-1)}$ can be updated normally and deviates from identity-like form. Therefore, $\frac{\partial \mathcal{L}}{\partial x^{(L-1)}}$ can get normal updating in the next step. As a result, all weights can be updated normally.

If $\forall i, D_{i+1} > D_i$, then the variance of gradients will increase from layer to layer. However, this will not make trouble for training. For $\frac{\partial \mathcal{L}}{\partial x^{(i)}}$, its variance $\sigma^2(\frac{\partial \mathcal{L}}{\partial x^{(i)}}) \approx \frac{D_{i+1}}{D_i}\sigma^2(\frac{\partial \mathcal{L}}{\partial x^{(i+1)}})$. Therefore, the variance of $\frac{\partial \mathcal{L}}{\partial x^{(0)}}$ can be calculated as

$$\sigma^2(\frac{\partial \mathcal{L}}{\partial x^{(0)}}) = \frac{D_1}{D_0}\frac{D_2}{D_1}\ldots\frac{D_L}{D_{L-1}}\sigma^2(\frac{\partial \mathcal{L}}{\partial x^{(L)}}) = \frac{D_L}{D_0}\sigma^2(\frac{\partial \mathcal{L}}{\partial x^{(L)}}). \qquad (15)$$

According to Eq. (15), the gradient will not explode in the backward procedure.

## A.3 IMPLEMENTING IDINIT ON ATTENTION LAYER IN TRANSFORMER

In this part, we show the way to initialize the attention layer with IDInit. Prior to that, formulating an attention layer as

$$\text{Att}(Q, K, V) = \text{softmax}(\frac{QW^QW^KK}{\sqrt{d}})VW^VW^O, \qquad (16)$$

where $Q$ is the query matrix, $K$ means the key matrix, $V$ denotes the value matrix, $W^Q$, $W^K$ and $W^V$ represents the weights for $Q$, $K$, and $V$ respectively, and $W^O$ is the output transformation. Following the instruction of IDInit in Sec. 3, we firstly use $\text{IDI}_\tau$ to initialize $W^Q$, $W^K$, $W^V$ and $W^O$. And then, we use $\text{IDIZ}_\varepsilon$ to initialize the last fully-connected layer $W^O$. The $\tau$ and $\varepsilon$ are consistently set with the paper content to 1 and 1e-6, respectively.

## B DETAILED SETTINGS OF EXPERIMENTS

In this paper, for ReLU activated networks, $\tau$ is set to $\sqrt{2}$ for the first layer in a network and 1 for other $\text{IDI}_\tau$ / $\text{IDIC}_\tau$ initializing layers, while for tanh-activated networks, all $\text{IDI}_\tau$ is set to 1, and $\varepsilon$ is $1e-6$ for all $\text{IDIZ}_\varepsilon$ / $\text{IDIZC}_\varepsilon$ initializing layers.

### B.1 DETAILS OF VALIDATION ON NON-RESIDUAL CONVOLUTION EXPERIMENT

In this experiment, we use AllConv (Springenberg et al., 2015) which consists of nine convolutional layers as the backbone network. We show the structure of AllConv in Table 4. The dataset is Cifar10. The optimizer is Stochastic Gradient Descent (SGD) with momentum 0.9, weight decay 5e-4, and learning rate 1e-1. The learning rate scheduler adopts a warm-up cosine reduction strategy. We run the model in 300 epochs on one Nvidia A100. We adopt Kaiming initialization and IDInit w/o $\text{IDIC}_\tau$ initialization for comparison. Since there is no residual connection, we do not consider the $\text{IDIZC}_\varepsilon$ function in this experiment. For each initialization, we have run them with 0, 10, 20, 30, 40, 50, and 60 warm-up epochs. The experiment is conducted on one Nvidia A100.

Table 4: Architectures of the tensorial All-Conv networks. Window means the convolutional kernel window size. Channels indicate $\mathbf{c}_{in}$ and $\mathbf{c}_{out}$ of a standard convolutional kernel $\mathcal{C} \in \mathbb{R}^{\mathbf{c}_{in} \times \mathbf{c}_{out} \times k \times k}$. The avg pool denotes the average pooling operation.

| Layer | Window | Channels |
|-------|--------|----------|
| conv1 | 3×3 | 3× 96 |
| conv2 | 3×3 | 96× 96 |
| conv3 | 3×3 | 96× 96 |
| conv4 | 3×3 | 96× 192 |
| conv5 | 3×3 | 192× 192 |
| conv6 | 3×3 | 192× 192 |
| conv7 | 3×3 | 192× 192 |
| conv8 | 1×1 | 192× 192 |
| conv9 | 1×1 | 192× 10 
 avg pool |

## B.2 DETAILS OF VALIDATION ON RESIDUAL CONVOLUTION EXPERIMENT

In this experiment, we validate the proposed initialization with the comparison with existing initialization, including (1) Fixup; (2) SkipInit; (3) ReZero; (4) Kaiming; (5) Zero $\gamma$ (Setting the scale in Batch Normalization (BN) to 0). We use ResNet-56/110 as backbones on Cifar10. To verify the universality, we use two settings, namely w/ and w/o BN. For analyzing convergence, we adopt both SGD and Adam optimizer for updating models. We set SGD, with the momentum 0.9, the weight decay 5e-4, and the learning rate 0.2. For Adam, the learning rate is 0.001, $\beta_1$ is 0.9 and $\beta_2$ is 0.999. We train models for 200 epochs. The learning rate is reduced with a cosine function. The experiment is conducted on one Nvidia A100.

## B.3 DETAILS OF ABLATION EXPERIMENT

The dataset is Cifar10 and the backbone is ResNet-20. We choose SGD with momentum 0.9, weight decay 5e-4, and learning rate 0.1 to train the models for 200 epochs. The learning rate is reduced with a cosine function. And data-augment mixup is applied. The experiment is conducted on one Nvidia A100.

## B.4 DETAILS OF DETERMINACY ANALYSIS ON TEXT CLASSIFICATION EXPERIMENT

We also explore performance networks on text classification datasets including SST2 and SST5 (Socher et al., 2013) and we select TextCNN (Kim, 2014), TextRNN (Lai et al., 2015) and Transformer (Vaswani et al., 2017) for comparison. For TextCNN and TextRNN, we use AdaDelta (Zeiler, 2012) optimizer with a learning rate 1.0 and adopt Adam (Kingma & Ba, 2015) for Transformer with a learning rate 1e-4. For the embedding layer, we utilize Glove (Pennington et al., 2014) and Word2Vec(Mikolov et al., 2013) to initialize the embedding weights. All models are trained up to 10 epochs. Fixing other hyper-parameters except for the initialization, we run all the random initialization for 5 times. The experiment is conducted on one Nvidia A100.

## B.5 DETAILS OF IMAGE CLASSIFICATION ON IMAGENET EXPERIMENT

In this experiment, we use ImageNet for validation. We use ViT-B/32 (Dosovitskiy et al., 2021), ResNet-50/152 (RN-50/152) and Se-ResNet-50 (SRN-50) as backbones. For ViT-B/32 that inputs $32 \times 32$ patch window, the optimizer is AdamW with a learning rate 1e-3 and a weight decay of 5e-2. And the batch size is 1024. The epoch for training is 300. We use 30 epochs for warm-up. The input image size is $224 \times 224$. The dropout rates of the embedding layer and the network layer are all 0.1. For RN-50/152 and SRN-50, the optimizer is SGD with a learning rate 1e-1 and a weight decay of 1e-4. And the batch size is 1024. The epoch for training is 90. We use 9 epochs for warm-up. The input image size is $160 \times 160$ for the front 35 epochs and $224 \times 224$ for the remaining epochs. For all models, we apply data-augment including cutmix (Yun et al., 2019) with $\alpha = 1.0$, mixup (Zhang

et al., 2018) with $\alpha = 0.8$, the switching probability is 0.5 and a label smoothing with 0.1. The experiment is conducted on 4 Nvidia A100.

### B.6 DETAILS OF PRE-TRAINING ON WUDAOCORPORA EXPERIMENTS

In this experiment, we conduct experiments on WuDaoCorpora (Yuan et al., 2021) which has 200 GB of training data and 72 billion Chinese characters. We use GPT with 8 and 96 layers as backbones. The optimizer is Adam with a learning rate 1e-3 and a weight decay of 5e-4. And the batch size is 1536. We use 20k steps for warm-up. For 8-layer GPT, we run 70k steps. For 96-layer GPT, we run 13k steps. The experiment is conducted on 32 Nvidia V100.

## C ADDITIONAL EXPERIMENTS

We provide additional experiments to further validate IDInit. $\tau$ and $\varepsilon$ are set the same as Sec. B.

### C.1 VALIDATION ON THE LINEAR STRUCTURE

This experiment is conducted on MNIST. We use five linear layers named Liner-5 whose hidden layers are all of dimension 512. The optimizer is SGD with momentum 0.9, weight decay 5e-4, and a learning rate 1e-1. The learning rate scheduler adopts a cosine reduction strategy. We run the model in 30 epochs on one Nvidia A100. We both consider Linear-5-tanh and Linear-5-ReLU which consist of Linear-5, and tanh and ReLU activation functions, respectively. The experiment is conducted on one Nvidia A100.

Table 5: Results of Linear-5 on MNIST. "Default" means the default initialization of models where Xavier is for Linear-5-tanh and Kaiming is adopted for Linear-5-ReLU.

| Init. | Linear-5-tanh | Linear-5-ReLU |
|---|---|---|
| Default | 98.26 | 98.21 |
| IDInit | **98.32** | **98.4** |

As shown in Table 5, IDInit can achieve the highest accuracy in both different tanh and ReLU conditions. The results show the ability of our proposed method to train a model with only fully-connected layers.

### C.2 ANALYSIS ON VARIANCE PROPAGATION

Here we conduct an experiment on Cifar10 to demonstrate data-flow will keep stable. We use 4 types of networks: (1) FC: 10-layer fully-connected layers; (2) ResFC: 10 residual blocks (two fully-connected layers in a block); (3) Conv: 9-layer AllConv in Sec. B.1; (4) ResConv: 10 residual blocks (two convolutional layers in a block). For (1) and (2) two fully-connected networks, we reshape Cifar10 data as $\mathbf{X} \in \mathbb{R}^{32 \times 96}$ as input and does not use any activation function. For (1), hidden lengths are $\{200, 400, 600, 800, 1000, 1000, 800, 600, 400, 200\}$. For (2), hidden lengths are all set to 96. For (3) and (4) two convolution networks, we directly input images to them, and use ReLU as the activation function. For (3), we directly use AllConv as shown in Table 4. For (4), we first use convolution to transfer an image to 16 channels, and then set the channels of all convolution within residual blocks to 16. For comparison, we use Xavier for (1) and (2), and Kaiming for (3) and (4) in terms of the activation function. We also employ noises with 0 mean, and $\{0.00, 0.01, 0.10, 1.00\}$ for comparing robustness. In the experiment, we run 500 rounds for each model. The experiment is conducted on one Nvidia A100.

Results are shown in Figure 10. The regular methods Xavier and Kaiming can only work on non-residual networks. On residual networks, they both cause giant standard derivation, leading to instability. By contrast, the proposed IDInit can consistently transit data-flow in an appropriate scale

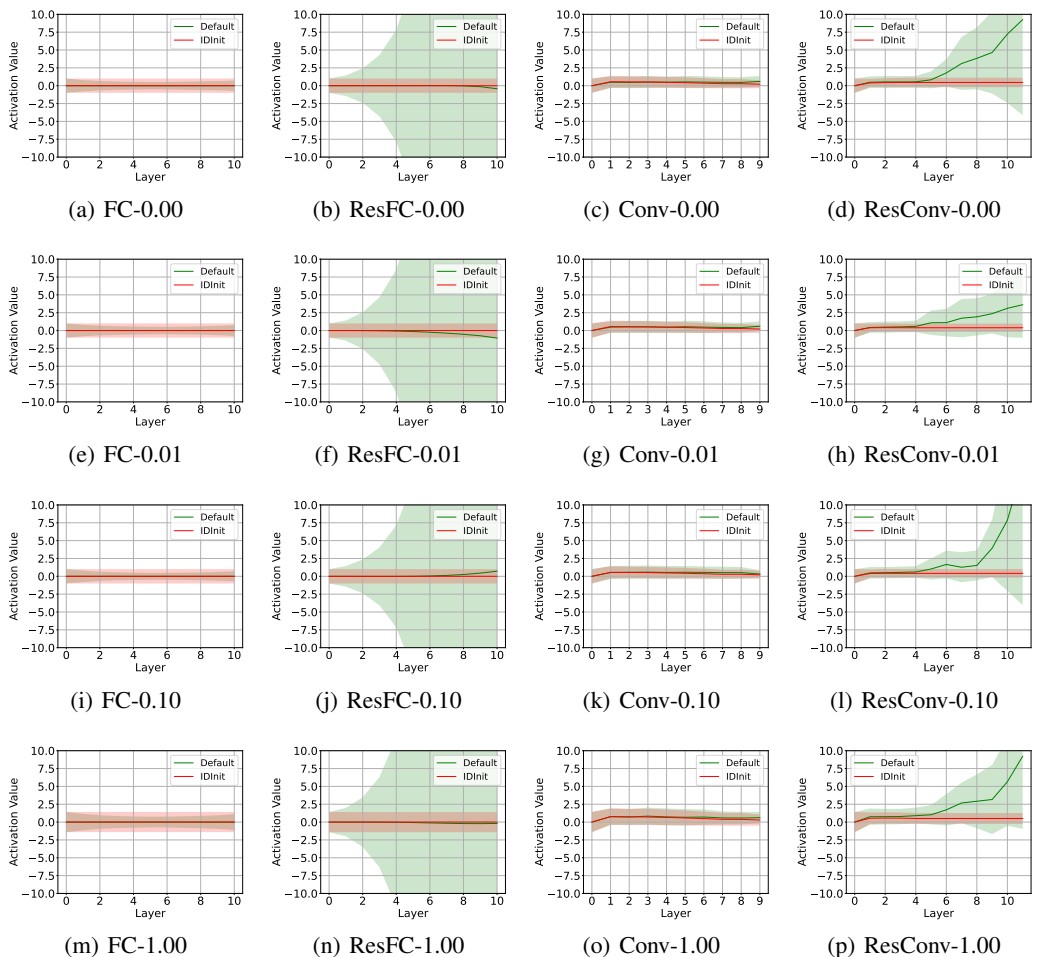

Figure 10: Results of the analysis on variance propagation. The numerical value after the model name means the standard derivation of the noise. "Default" means the default initialization of models, specifically, Xavier for FC and ResFC, and Kaiming for Conv and ResConv. The default methods can only work on non-residual networks FC and Conv, however, fail on residual networks ResFc and ResConv, for cause instability with giant standard derivation. By contrast, IDInit can consistently transit data-flow in an appropriate scale on all models and various noises, which shows sufficient robustness, and can provide models with stable and efficient training.

on all models and various noises, which shows sufficient robustness, and can provide models with stable and efficient training.

### C.3    ANALYSIS ON WEIGHT DISTRIBUTION

In this experiment, we conduct an experiment on Cifar10 with ResNet-20 to show the weight distribution of IDInit. We use an SGD optimizer with a learning rate 0.2, and weight decay 5e-4. The batch size is 1024. Training epochs are 200. The learning rate is reduced with a cosine function. The experiment is conducted on one Nvidia A100.

The results are shown in Figure 11, weights initialized with IDInit are almost full of zero at the beginning, while Kaiming uses a Gaussian distribution. At the end of the training, IDInit still contains more zero values than Kaiming, which is beneficial for memory occupation since a 0 value will not cost memory space.

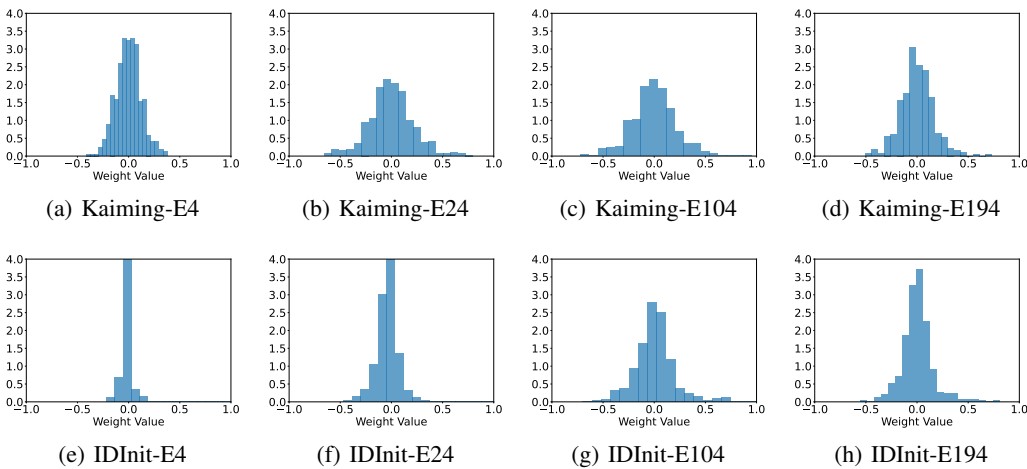

Figure 11: Histograms of the first convolution weights in ResNet-20. "E" means the epoch index. IDInit contains more zero values in each epoch compared with Kaiming initialization.

### C.4 ANALYSIS ON INPUT-OUTPUT JACOBIAN

Here we conduct an experiment on Cifar10 with ResNet-130 to demonstrate IDInit follows the dynamical isometry. We remove batch normalization for the more clear difference between IDInit and Kaiming. We use an Adagrad optimizer with a learning rate 0.1. The batch size is 100. The activation is ReLU. The experiment is conducted on one Nvidia A100.

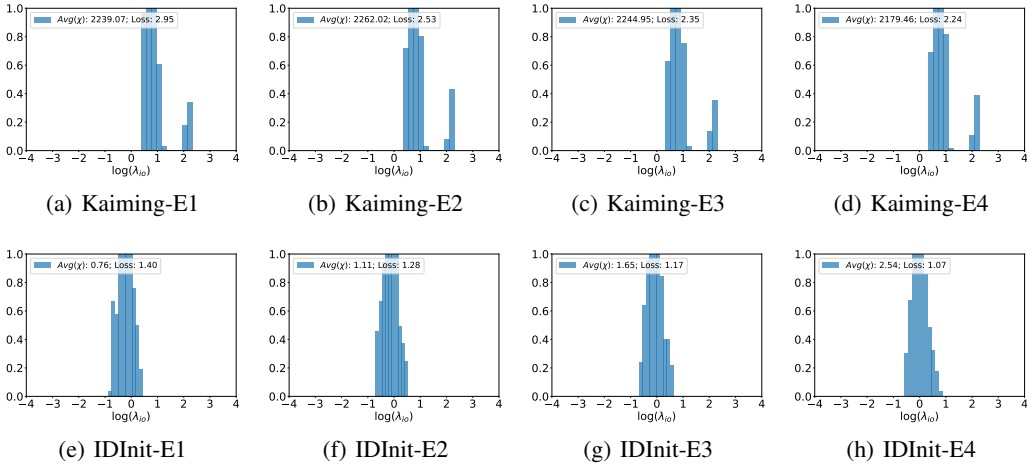

Figure 12: Histograms of log singular values ($\log(\lambda_{io})$) for the input-output Jacobian. "E" means the epoch index. Compared with Kaiming initialization, IDInit has a significantly smaller squared singular value $\chi$, which can achieve a faster reduction of the loss.

As shown in Figure 12, Kaiming initialization cause a high squared singular value $\chi$, reaching more than 2000. Compared to Kaiming, IDInit only derives $\chi$ around 1, indicating correspondence to the dynamical isometry. In addition, the loss of IDInit decreases faster than Kaiming, which shows a good convergent ability.

