# OpenReview forum: "Identical Initialization: A Universal  Approach to Fast and Stable Training of Neural Networks"
_ICLR.cc/2023/Conference — Submitted to ICLR 2023_

### Official Review · Reviewer_st9W · 2022-10-24

**Confidence:** 3
**Clarity, Quality, Novelty And Reproducibility:** Clarity is fine. Novelty is close to …
**Correctness:** 2
**Technical Novelty And Significance:** 1
**Empirical Novelty And Significance:** 1
**Recommendation:** 3

**Strength And Weaknesses:**

Strengths:
1. The experiments are done on many datasets.
2. The code is attached for reproducibility.

Weaknesses:
1. It has been shown (Thereom 5 in [1]) that the identity initialization may not lead to convergence for deep linear neural networks.
2. For residual layer, this work proposes to initialize the weight matrix with tiny values so as to avoid the dead neuron problem. This is not sensible. Because of the shortcut connections, weights with zero initialization still receive non-zero gradients.
3. The overall idea, identity initialization, is not novel. For example, see [2]. The proposed variant in this paper is technically trivial.

References:
[1] Gradient descent with identity initialization efficiently learns positive definite linear transformations by deep residual networks, Peter L. Bartlett, David P. Helmbold, Philip M. Long, ICML 2018.
[2] A Simple Way to Initialize Recurrent Networks of Rectified Linear Units, Quoc V. Le, Navdeep Jaitly, Geoffrey E. Hinton, arXiv 2005.



**Summary Of The Paper:**

This paper proposes to initialize each linear layer of a deep neural network to be identity so as to accelerate training.

**Summary Of The Review:**

This paper proposes identity initialization for deep neural networks to accelerate the training. I found the idea lack of novelty and the claim of solving the dead neuron problem insensible.

---

> ### Author Response · Authors · 2022-11-19
> **Response to reviewer st9W**
>
> Thanks for the time of the reviewer to review our paper.
>
> >It has been shown (Thereom 5 in [1]) that the identity initialization may not lead to convergence for deep linear neural networks.
>
> Theorem 5 may not hold in practice. We have written a paragraph in the original paper to discuss with [1] for the close relation to the proposed IDInit. We had noticed the Thereom 5 indicating a failure to converge. However, as we claimed that there is a "gap between reality and theory". [1] is derived based on multiple linear layers. [1] uses a strong assumption that a residual net is equivalent to multiple linear layers if the weights are small. However, value 1 in the identity matrix is not small. Therefore, the conclusion from [1] may not be suitable for non-linearity and residual connection. In addition, [1] also uses simple gradient descent updates without considering auxiliary techniques like momentum and weight decay and so on. Therefore, this Thereom 5 may not hold in practice. Moreover, our empirical results show good improvement compared to previous studies, which can also demonstrate the reasonability for applying an identity matrix as an initialization ingredient. As so far, there is not a convincing phenomenon to support Thereom 5 in practice.
>
> >For residual layer, this work proposes to initialize the weight matrix with tiny values so as to avoid the dead neuron problem. This is not sensible. Because of the shortcut connections, weights with zero initialization still receive non-zero gradients.
>
> To explain this problem, we would like to use a simplified residual block considering the downsampling operation and batch normalization. This block can be formulated as
> $$
> x_{i+1} = \gamma((u+w^{(0)}w^{(1)})x_i),
> $$
> where $\gamma$ is the multiplier of batch normalization, $u$ denote a residual connection. All the variable is scalar. In a practical case of ResNet-20 (implementation URL: https://github.com/hongyi-zhang/Fixup/blob/master/cifar/models/resnet_cifar.py), $u=0$ when the block is a downsampling block, and $u=1$ is for a non-downsampling block. The gradient of $w^{(0)}$ is
> $$
> \frac{\partial L}{\partial w^{(0)}} = \frac{\partial L}{\partial x_{i+1}}\frac{\partial x_{(i+1)}}{\partial w^{(0)}}=\frac{\partial L}{\partial x_{i+1}}(\gamma w^{(1)}x_i).
> $$
> If setting $w^{(1)}=1$ and $w^{(0)}=0$, when $\gamma=0$, $\frac{\partial L}{\partial w^{(0)}}=0$. Even in a $\gamma=1$ case, if a non-downsampling block follows a downsampling block, then $x_i=0$, causing $\frac{\partial L}{\partial w^{(0)}}=0$. Therefore, setting $w^{(0)}=0$ often causes a 0 gradient known as a dead neuron problem. Therefore, the dead neuron problem is sensible, and IDIZ is reasonable for solving this problem.
>
> >The overall idea, identity initialization, is not novel. For example, see [2]. The proposed variant in this paper is technically trivial.
>
> We respectably disagree with this conclusion. The reviewer seems to cite the wrong year of [2] that is preprinted in 2015. [2] is specifically designed for RNNs, and only sets the hidden-to-hidden matrix to an identity matrix. In detail, a vanilla RNN layer can be formulated as
> $$
> y = W{x}+H{h},
> $$
> where $x\in \mathbb{R}^m$ is an input, $h\in \mathbb{R}^n$ is a hidden state, $h\in \mathbb{R}^n$ is an output,  $W\in \mathbb{R}^{n\times m}$ is an input-to-hidden matrix, and $H\in \mathbb{R}^{n\times n}$ is a hidden-to-hidden matrix. [2] only set the $H$ to an identity matrix. By contrast, IDInit set both $W$ and $H$ to identity matrices. Moreover, [2] has not considered a non-square situation, which is a key point for universal use. And [2] also has not considered the convolution condition, which is significantly different from the proposed IDInit. For more related studies mentioned by other reviewers, i.e., ZerO in Wd9d, Fixup in TjT6, and  DiracNet and ISONet in QUve, we have also detailly discussed them. We sincerely hope the reviewer can reconsider our contribution and novelty.

---

### Official Review · Reviewer_QUve · 2022-10-24

**Confidence:** 5
**Correctness:** 2
**Technical Novelty And Significance:** 2
**Empirical Novelty And Significance:** 2
**Recommendation:** 3

**Clarity, Quality, Novelty And Reproducibility:**

The clarity of this paper can be improved a lot. I listed a few examples in the second point of the previous weakness section. I think the author should address them and further polish the storytelling and presentation. Besides, I feel I don’t obtain much insight from reading this paper. The authors may also consider to visualize the distribution of the weight matrix, e.g. how the weight matrix value changes through the training.

The paper has its own novel part. However, the central identity initialization has been proposed an analyzed before (ISONet and DIractNet, listed in the weakness).

It provides code. I checked the relevant implementation. I think the proposed method is simple and also easy to reproduce.

**Details Of Ethics Concerns:**

Not Applicable.

**Strength And Weaknesses:**

Strength:
1) This paper is motivated by the dynamic isometry formulation and propose a very simple initialization (identity initialization). The simplicity is good and empirically helpful on varies settings.
2) The proposed two techniques on improving the feature diversity are interesting.

Weakness:
1) Missing Reference: This paper misses discussion with DiracNet [1] and ISONet [2]. DiracNet reparameterize the convolutional kernals and essentially initialize it to be an identity. ISONet explicitly initialize the conv kernels to be identity both in vanilla convnets and residual convnets. It is necessary to discuss the differences and advantages over these previous works.
2) Some terms and examples are confusing.
- the robustness in this paper actually means insensitivity to hyperparameters. However, this term in machine learning usually means the network is insensitive to input noise.
- Table 1 is also confusing. Most of the entries in the “w/ BN” category should be put in the “w/o” because Zero \gamma, Fixup, Skip Init, ReZero are designed for training residual networks without normalization.
- Figure 7 can be replaced by a Table becuase the current figure does not have caption and it is not straightforward to get the point from the figure.
- The experiment setting and conclusion of section 4.4 is also confusing. Is the proposed initialization has any randomness? It seems the initialization is deterministic. In this case, why the std is not 0 in Table 1?
3) The central message of the paper seems to be unclear. The authors mix two information together: 1) the trainability of network without batch norm; 2) the proposed method improves over the original initialization. The first part is well-motivated because BN has many disadvantages. However, the proposed method only shows the trainability of a network with 100 layers on CIFAR. The ImageNet experiments are based on networks with varies normalization layers and the proposed method only has marginal improvement over the baselines.

[1] Zagoruyko et al. DiracNets: Training Very Deep Neural Networks Without Skip-Connections.
[2] Qi et al. Deep Isometric Learning for Visual Recognition. ICML 2020.

**Summary Of The Paper:**

This paper proposes a new initialization scheme for both vanilla convnets and convnets with residual structures. This initialization is a simple and special case to maintain dynamic isometry at initialization time. Specifically, they initialize the weight matrix to be identity. The authors also propose a spatial shift scheme to improve the representation learning ability. Using this initialization, they can train nine-layer vanilla convnets and 110-layer residual networks on CIFAR. It is also shown this initialization method is helpful for transformer training.

**Summary Of The Review:**

In summary, this paper is interesting as it shows simply initializing the kernels to be identity can make vanilla networks trainble. However, the writing, experiments, and presentation need to be improved a lot. I cannot recommend acceptance for this paper before this improvement.

---

> ### Author Response · Authors · 2022-11-19
> **Response to reviewer QUve (Part 1)**
>
> Thanks to the reviewer to give a thoughtful review of our paper.
>
> >(1) Missing Reference: This paper misses discussion with DiracNet [1] and ISONet [2].
> >
> >(2) The paper has its own novel part. However, the central identity initialization has been proposed an analyzed before (ISONet and DIractNet, listed in the weakness).
>
> We will add a discussion on DiracNet [1] and ISONet [2]. DiracNet maintains an identity for propagating information deeper into the network. However, it suffers from reducing residual connection, causing performance loss. Compared with it, IDInit can simply be used in residual nets for better performance. As for ISONet, it is an isometric learning framework that contains an identical initialization (i.e., the Dirac function that is also used in ZerO by padding 0 in a non-square matrix case), and isometric regulation in training. In a residual net, ISONet multiplies 0 to the residual stem like Fixup. By contrast, IDInit set the last weight to 0 without any more parameters. ISONet surely lacks the flexibility for various convolutions as it specifies the net without normalization, and requires SReLU. And it has the potential to encounter rank constraint problems mentioned in ZerO. Only using the initialization of ISONet with their regulation on weights will cause awful results as shown in Table 3 in response of Reviewer Wd9d. Moreover, ISONet is not applicable to the Transformer as its activation function is Gelu. Compared to that, IDInit relives rank constraint inherently, avoids dead neural problems, and can be widely applicable.
>
> >The robustness in this paper actually means insensitivity to hyperparameters. However, this term in machine learning usually means the network is insensitive to input noise.
>
> We have changed the robustness to stability or universality accordingly.
>
> >Table 1 is also confusing. Most of the entries in the “w/ BN” category should be put in the “w/o” because Zero \gamma, Fixup, Skip Init, ReZero are designed for training residual networks without normalization.
>
> We have changed "w/ BN" to "w/ multiplier", and "w/o BN" to "w/o multiplier". We would like to note that Zero $\gamma$, and ReZero contain batch normalization in their structures, and SkipInit and Fixup remove the batch normalization. However, SkipInit and Fixup still have a multiplier, they can inherently derive better results than a ResNet w/o BN. Therefore, we compare with SkipInit and Fixup in a ResNet w/ BN condition for a fair comparison. As w/ BN is not rigorous for SkipInit and Fixup, we change "BN" in Table 1 to "multiplier".
>
> >Figure 7 can be replaced by a Table becuase the current figure does not have caption and it is not straightforward to get the point from the figure.
>
> We have replaced Figure 7 in the original paper with a table in the revision.
>
>
> >The experiment setting and conclusion of section 4.4 is also confusing. Is the proposed initialization has any randomness? It seems the initialization is deterministic. In this case, why the std is not 0 in Table 1?
>
> The experiments of Table 1 in the original paper are conducted in a random training framework which includes random batch sequence, random data crop, random data flip, and so on. Therefore, the std is not 0 in Table 1. By contrast, in Sec. 4.4, to analyze determinacy from initialization, we conduct experiments by fixing all random factors including random batch sequence, random data crop, random data flip, and so on. As a result, the std is 0 in Table 2 in the original paper.

---

> > ### Author Response · Authors · 2022-11-19
> > **Response to reviewer QUve (Part 2)**
> >
> > >The central message of the paper seems to be unclear. The authors mix two information together: 1) the trainability of network without batch norm; 2) the proposed method improves over the original initialization. The first part is well-motivated because BN has many disadvantages. However, the proposed method only shows the trainability of a network with 100 layers on CIFAR. The ImageNet experiments are based on networks with varies normalization layers and the proposed method only has marginal improvement over the baselines.
> >
> > Please refer to [R1] in the response of Wd9d for details of IDInit. We have elaborated on "The Whole Design of IDInit" there, including the central message. We aim to develop a universal initialization to diverse neural networks for faster convergence, higher stability, and better performance, therefore, ResNets w/ and w/o BN both belong to our consideration.
> >
> > 1) Networks without batch norms are in tough conditions for training. Trainability on networks without batch norms can show good stability. We conduct an experiment on Cifar-10 to validate the stability on ResNet without batch norm. As IDInit is the only initialization, we will compare IDInit with the default initialization Kaiming. The optimizer is SGD with weight decay 5e-4 and learning rate 1e-1. The batch size is 128. We warm up for 10 epochs.
> >
> > Table 8. Results of ResNets without batch norm on Cifar-10 at Epoch 20.
> > | Initialization   | ResNet-100 | ResNet-500 | ResNet-1202 |
> > | ------- |:---------:|:---------:|:----------:|
> > | Kaiming |   10.00   |   10.00   |   10.00    |
> > | IDInit  |   71.74   |   73.71   |   75.38    |
> >
> > Results show that Kaiming fails in all models. By contrast, IDInit successfully trains ResNet without batch norm from 100-layer to 1202-layer, indicating a good stable ability.
> >
> > 2) The improvement of IDInit on the original initialization lies on multiple aspects: (a) faster convergence: IDInit converges significantly faster than the original initialization; (b) higher stability: the original initialization of ResNet is Kaiming initialization which cannot stabilize the ResNets without BN, while IDInit can successfully train ResNets without BN; (c) better performance: although IDInit have marginally performance improvement in the ImageNet experiment, the improvement of IDInit is consistently better. In other experiments, an improvement in the performance of IDInit is more significant. For example, in Table 2 of the original paper, for the performance of $TextRNN_G$ on SST5, IDInit can achieve a 2.85% improvement in accuracy compared to the default initialization, which is sufficiently significant. Therefore, we believe the improvement from IDInit is sufficiently significant compared to the original initialization.
> >
> >
> >
> > >Besides, I feel I don’t obtain much insight from reading this paper. The authors may also consider to visualize the distribution of the weight matrix, e.g. how the weight matrix value changes through the training.
> >
> > For adding insights, we have illustrated variance propagation in Figure 11 in the appendix of the original paper. We also add an analysis of rank constraints and replication problems in Table 1 and Table 2 of response to Wd9d, respectively. Moreover, we add analyses on the weight distribution and input-output Jacobian in Sec. C.3 and Sec. C.4 in the appendix of the revision, respectively. We hope these modifications can resolve the concern of the reviewer.

---

> > > ### Comment · Reviewer_QUve · 2022-11-20
> > > **Response**
> > >
> > > I thank the authors for their detailed response and incorporation of my feedback. I hope the authors can find them helpful. The authors include the missing reference, correct some terms, and add further visualizaiton of the proposed method. However, there are still questionable parts which make me keep my original recommendated score:
> > >
> > > I agree “Networks without batch norms are in tough conditions for training.”, but I don’t agree why the authors claim IDInit is the only one that achieves training without batchnorm. Fixup, ISONet, and ReZero can also do that. The claimed faster converence and better performance is marginal.
> > >
> > > I don’t think the randomness experiments in Table 3 make sense. If the authors fix other random factor but only keep the initialization (sampling matrix weights from a gaussian) random, it would be obvious that the proposed method will have 0 variance. However, I don’t think this can be an evidence of any further claims.

---

> > > > ### Author Response · Authors · 2022-11-25
> > > > **Reply to Reviewer QUve (continued)**
> > > >
> > > > We appreciate the prompt feedback from the reviewer.
> > > >
> > > > >I agree “Networks without batch norms are in tough conditions for training.”, but I don’t agree why the authors claim IDInit is the only one that achieves training without batchnorm. Fixup, ISONet, and ReZero can also do that. The claimed faster converence and better performance is marginal.
> > > >
> > > > Sorry for the confusion. Indeed, "ResNet without batch norms" in our paper denotes the category of variants of ResNets whose batch norms are eliminated without extra manipulation. To avoid this confusion, we replace "ResNet without batch norms" in Table 1 in the original paper as "ResNet without multiplier" in the revision. Thus under this definition, Fixup and ReZero do not belong to this category as they both add extra multiplier parameters. In addition, ISONet incorporates a Dirac initialization,  as well as the SReLU activation function and an orthogonal regulation. Without the orthogonal regulation, Dirac initialization can cause a fatal performance, which can break down the training of ResNet, as shown in the following Table 9.
> > > >
> > > > Table 9. Performance in AllConv on Cifar-10 (The original table is Table 3 in response of Reviewer Wd9d. Please refer to there for more details.). The zero-padding convolution is exactly the initialization of ISONet. ISONet cannot be trained well without additional regulation on weights, and we will not apply such special regulation as the regulation is not a common setting.
> > > > | Model            |     Accuracy     |
> > > > | ---------------- |:----------------:|
> > > > | zero-padding (Dirac initilization)    | 23.64 $\pm$ 0.55 |
> > > > | identity-padding (IDInit w/o IDIC) | 74.94 $\pm$ 0.91 |
> > > >
> > > > Indeed, the faster convergence and better performance are significant. We would like to detail the results on the ImageNet dataset originally shown in Figure 7 into Table 10.
> > > >
> > > > Table 10. Results on ImageNet. The average increment is 0.55% on the baseline, which is a significant improvement. The value in brackets means "Epochs to 60% Acc.".
> > > >
> > > > | Model   |  ViT-B/32  | RN-50 (Adamw) |   RN-50    |   SRN-50   |   RN-152   | Avg ($\Delta$) |
> > > > | ------- |:----------:|:-------------:|:----------:|:----------:|:----------:|:--------------:|
> > > > | Default | 71.05 (44) |  76.20 (20)   | 75.70 (38) | 76.30 (32) | 78.76 (28) |     0 (0)      |
> > > > | IDInit  | 71.60 (42) |  76.71 (14)   | 76.72 (24) | 76.93 (22) | 79.10 (23) |   0.55 (7.4)   |
> > > >
> > > > Table 10 shows that the faster convergence and performance are actually significant --- the average improvement is 0.55%, which is significantly larger than the improvement brought by ZerO[3]  (i.e., 0.03% as shown in Table 2 of [3] ).  As for the convergence, we have already shown significantly fewer "Epochs to 80% Acc." in Table 1 of the revision. This epoch metric is also used in ReZero[4] as a convergence indicator. And there is also an average of 7.4 epoch acceleration in the ImageNet experiment. We believe these metrics can demonstrate IDInit is a sufficiently good initialization method.
> > > >
> > > > >I don’t think the randomness experiments in Table 3 make sense. If the authors fix other random factor but only keep the initialization (sampling matrix weights from a gaussian) random, it would be obvious that the proposed method will have 0 variance. However, I don’t think this can be an evidence of any further claims.
> > > >
> > > > We respectably disagree with the reviewer ignoring the significant performance of IDInit to claim this experiment does not make sense. As demonstrated in Table 3, fixing other random factors, IDInit has led to two clear advantages: (1) zero variance, and (2) significantly better performance (e.g., IDInit has achieved a 2.85% improvement for $TextRNN_G$ on SST5). Observing the above advantages, we claim that IDInit can be a promising component to design a high-performance deterministic framework to train a model without variance, which can be efficient for practitioners to derive a better-trained model without multiple trials. It can also benefit further theoretical analysis of the dynamics in training neural networks given stable and fixed initialization. Therefore, we believe that this experiment is reasonable and necessary.
> > > >
> > > >
> > > > [3] Jiawei, Zhao, Florian Schäfer, and Anima Anandkumar. "ZerO Initialization: Initializing Neural Networks with only Zeros and Ones."
> > > >
> > > > [4] Thomas, Bachlechner, et al. "Rezero is all you need: Fast convergence at large depth." Uncertainty in Artificial Intelligence. PMLR, 2021.

---

### Official Review · Reviewer_TjT6 · 2022-10-26

**Confidence:** 3
**Correctness:** 2
**Technical Novelty And Significance:** 2
**Empirical Novelty And Significance:** 3
**Recommendation:** 3

**Clarity, Quality, Novelty And Reproducibility:**

As mentioned above, the paper is poorly written, with many grammatical mistakes and vague language. This makes it difficult to discern the novelty of the proposed approach compared to existing ones. Given the authors' awareness of FIXUP and ZerO, the claim "To our best knowledge, it is the first trial to put identical-like initialization into practice" appears to be false.
Regarding the numerical experiments, I am somewhat concerned by the low performance of the baseline in Figure 6.

**Strength And Weaknesses:**

### Strength
The main strength of the paper are the empirical improvements apparently achieved by the proposed initialization scheme.

### Weakness
The first-order weakness of the paper is the poor quality of its writing. There are numerous grammatical mistakes and vague language such as "Therefore, if all values are set to 0, it can obtain non-zero gradients of a suitable magnitude." Due to the above, despite numerous attempts, I am still not entirely confident if I have correctly understood the approach proposed by the authors.

A second weakness (possibly related to the first) is that the motivation for the proposed approach over FIXUP and ZerO is not clear. In particular the fixup approach, that similarly initializes the residual branches as products of identities and zero seems closely related.


**Summary Of The Paper:**

The reviewed work proposes "identical initialization" as a novel initialization scheme for neural networks. The guiding principle of this initialization scheme is to ensure that each layer, at initialization, propagates activations identically.

The motivation behind this choice is that the resulting isommetry property of the network prevents vanishing or exploding gradients and thus allows for the stable training of deep architectures.

Numerical experiments on classification for CIFAR10 and Imagenet show improvements due to the new initialization scheme.

**Summary Of The Review:**

In summary, the empirical results suggest that the authors may be up to an interesting result. However, the writing, presentation, and comparison to existing work need major improvement. In its present state, the paper is not ready to be published at a major ML conference.

---

> ### Author Response · Authors · 2022-11-19
> **Response to reviewer TjT6**
>
> We thank the reviewer for the time to read our paper.
>
> >(1) The first-order weakness of the paper is the poor quality of its writing. There are numerous grammatical mistakes and vague language such as "Therefore, if all values are set to 0, it can obtain non-zero gradients of a suitable magnitude." Due to the above, despite numerous attempts, I am still not entirely confident if I have correctly understood the approach proposed by the authors.
> >
> >(2) As mentioned above, the paper is poorly written, with many grammatical mistakes and vague language. This makes it difficult to discern the novelty of the proposed approach compared to existing ones.
> >
> >(3) A second weakness (possibly related to the first) is that the motivation for the proposed approach over FIXUP and ZerO is not clear. In particular the fixup approach, that similarly initializes the residual branches as products of identities and zero seems closely related.
>
> We have carefully checked typos in the original paper, and revised the sentence "Therefore, if all values are set to 0, it can obtain non-zero gradients of a suitable magnitude". In addition, we have also revised the writing suggestion from Reviewer QUve. Please check the revision.
>
> Please refer to [R1] "The Whole Design of IDInit" in response of Wd9d for motivation details. Here, we will provide a short elaboration.
> IDInit mainly aims for universal and flexible use in wide-range networks, as most initialization are limited for applicability. For example, Fixup and ZerO are proposed for residual nets, and ReZero requires a modification of architectures. In this paper, we design IDInit by considering fast convergence, high performance, and stable ability as important characteristics while keeping universal applicability.
>
> For the difference among IDinit, FIXUP, and ZerO, we would like to first formulate a residual layer as
> $$
> y = BN((1+\theta^{(0)}\theta^{(1)})x),
> $$
> where BN means batch normalization. Fixup set $\theta^{(1)}$ with common initialization like Kaiming, set $\theta^{(0)}$ to 0, and replace BN with a multiplier. And Fixup only supports residual nets. Therefore, Fixup is different from IDInit. As for ZerO, we detailly compare it with IDInit in [R1]. Please refer to [R1] for details. We believe there is the sufficient novelty of IDInit over prior studies.
>
> > Given the authors' awareness of FIXUP and ZerO, the claim "To our best knowledge, it is the first trial to put identical-like initialization into practice" appears to be false.
>
> The "identical-like initialization" means the initialization comes from a simple transformation of the identity matrix, rather than only considering identity transition. Fixup uses a common initialization, e.g., Kaiming initialization, thereby not an identity-like initialization. ZerO uses the Hadamard matrix to change identity-like matrices, causing a loss of identity. As for IDInit, our identity-padding matrix mentioned in response of Wd9d is identity-like, IDIC only reshapes identity-padding matrices, and IDIZ can be regarded as a concatenation of two identity matrices. Therefore, we claim IDInit is the first trial to put identical-like initialization into practice.
>
> >Regarding the numerical experiments, I am somewhat concerned by the low performance of the baseline in Figure 6.
>
> Kaiming shows a normal accuracy, and we also give mean $\pm$ std results of Kaiming in Table 3 of the response of Reviewer Wd9d. And IDInit w/o IDIC performs worse. We interpreted this phenomenon in [R1S3] of the response of Reviewer Wd9d.

---

> > ### Comment · Reviewer_TjT6 · 2022-12-03
> > **Thank you for your response**
> >
> > Thank you for your response. I recommend you continue to work on improving the paper to explain clearly the relationship to existing work. After looking at the other reviews, I believe that this paper should undergo another full round of review and should therefore be rejected this time.

---

### Official Review · Reviewer_Wd9d · 2022-10-29

**Confidence:** 5
**Correctness:** 2
**Technical Novelty And Significance:** 3
**Empirical Novelty And Significance:** 3
**Recommendation:** 3

**Clarity, Quality, Novelty And Reproducibility:**

The paper is mostly clear with high quality. However, I believe it lacks enough novelty. Identical initialization has been studied in ZerO paper [1], where IDInit and the proposed variant (Eq 14) in convolution are already considered. Its deterministic benefit is also studied in ZerO. It would be great if the authors can provide more details about the connection and difference between the two methods. Another concern is the reproducibility given the training degeneracy issue discussed earlier, which affects the effectiveness of the proposed method.

**Strength And Weaknesses:**

The paper is well-written and easy to follow. The authors also present extensive empirical evaluations to support their methods.

However, I have a severe concern regarding the effectiveness of IDInit. The identical initialization has been well studied in ZerO paper [1], where the authors theoretically prove that a simple identical initialization leads to training degeneracy in practical networks. The major reason behind this training degeneracy is that for a dimension-increasing matrix (e.g., the $D_{i+1} > D_{i}$ case in the current paper), a simple "identity + zero-padding" causes the rank constraint ((ii) in Theorem 1 in ZerO).

To initialize dimension-increasing matrix, IDInit replicates the inputs to span over the output dimension. However, this causes a similar rank constraint (i.e., a symmetric problem) as each replica in the dimension-increasing matrix receives the same gradient (excludes some special cases). ConstNet paper [2] also shows the degeneration caused by insufficient feature diversity (i.e., the symmetry), which is related to the replication here. In addition, the authors only perform trainability analysis from a signal propagation perspective, without considering the potential degradation caused by the replication.

I highly recommend the authors to empirically evaluate if IDInit can pass the simple rank verification in Figure 3 of ZerO paper. On the other hand, I suspect the surprisingly good empirical evaluation of IDInit comes from randomized operations (e.g., dropout) in networks, which directly breaks the training degeneracy. I suggest the authors to carefully consider this training degeneracy issue, as it affects the effectiveness and universality of the proposed method.

Other comments:

1. The need of zero preserving initialization (IDIZ) is unclear. The authors motivate IDIZ by arguing that Fixup or ZerO method will not work if the multiplier in batch normalization is initialized as 0. However, this is not a standard case because the multiplier in BN is usually initialized as 1. The motivation also requires a special case where BN is applied after the last conv layer. These reasons can not convince me why IDIZ is really needed. Do authors initialize the multiplier in BN as 0 in their empirical evaluation? If so, I suggest the authors to provide more details about this case.

2. How does IDIC compare to the simple IDI in convolution case? The authors mention that IDIC is used to improve feature diversity but it's unclear why it is needed. ZerO paper [1] directly utilizes IDI in convolution (Algorithm 2 in ZerO) and it works well.

3. Comparison to ZerO. Since ZerO is the most relevant work to IDInit I believe, I suggest the authors to empirically compare them, including both performance and determinacy analysis (since both methods claim the benefit as being deterministic).

[1] https://arxiv.org/abs/2110.12661

[2] https://proceedings.mlr.press/v119/blumenfeld20a.html

**Summary Of The Paper:**

The authors propose Identical Initialization (IDInit) that initializes weights using identity matrix and its variants. They consider how to initialize non-square matrices, residual structures and convolutional operations using identity-like methods. Empirical evaluation demonstrates its good performance and fast convergence.

**Summary Of The Review:**

Identical initialization is a very interesting topic and IDInit looks promising. My major concerns are its effectiveness (training degeneracy problem), necessity (motivations behind IDIZ and IDIC) and novelty (compared to ZerO). I would like to discuss these issues with the authors, and I am happy to raise my score if the authors can address my concerns.

---

> ### Author Response · Authors · 2022-11-19
> **Response to reviewer Wd9d (Part 1)**
>
> Thanks for the reviewer's hard work on our paper.
>
> **[R1] The Whole Design of IDInit**
>
> Since the reviewer is concerned about the effectiveness and motivation of IDIC and IDIZ, we would like to elaborate on the whole design of IDInit firstly.
>
> IDInit mainly aims for universal and flexible use in wide-range networks, as most initialization are limited for applicability. For example, Fixup and ZerO are proposed for residual nets, and ReZero requires a modification of architectures. In this paper, we design IDInit by considering fast convergence, high performance, and stable ability as important characteristics while keeping universal applicability.
>
> **[R1S1]** Motivation on Identity Matrix. We have introduced the identity matrix that can help convergence, performance, and stability by following the isometric mechanism as in Sec. 1.
>
> **[R1S2]** Motivation on Identity-Padding in  Dimension-Increasing Matrix. We implement an identity-padding matrix that can naturally solve the rank constraint problem. In detail, ZerO pads 0 values. According to A.2 Additional Proofs of (ii) in [1], the padded zeros cause zero gradients, leading to the rank constraint problem. By contrast, our identity-padding matrix can always derive non-zero gradients, thereby, being able to be trained without rank constraint. We re-implement simple rank verification in Figure 3 of the ZerO paper as the reviewer recommends. We choose a hidden dimension of 2048, and the input dimension is 768.
>
> Table 1. Verification of rank constraint.
> | Model            | Epoch 1 | Epoch 2 | Epoch 3 | Epoch 15 | Epoch 20 |
> | ---------------- |:-------:|:-------:|:-------:|:--------:|:--------:|
> | zero-padding     |    0    |   10    |   724   |   748    |   750    |
> | identity-padding |    0    |   100   |  1462   |   1544   |   1635   |
>
> As shown in Table 1 in the rebuttal. The zero-padding causes a rank constraint of 768. We also validate these two padding strategies in the AllConv experiment. We use Eq. (14) to transform the two kinds of matrices into convolutions. To reduce the influence of randomness, we use a "deterministic" setting in Pytorch and remove dropout in AllConv. Other training settings follow Sec. 4.1 in the original paper. The correlation is shown in Table 2, and the performance comparison is in Table 3.
>
> Table 2. Correlation in trained models.
> | Model            | Forward | Backward |
> | ---------------- |:-------:|:--------:|
> | Kaiming          |  5.13   |   1.07   |
> | zero-padding     |  0.00   |   0.64   |
> | identity-padding |  6.86   |   0.98   |
>
>
> Table 3. Performance in AllConv. The zero-padding convolution is also used in ISONet mentioned by Reviewer QUve. ISONet can be trained well for additional regulation on weights, and we will not apply such special regulation as we use a common setting. The big variance of Kaiming is caused by training instability as also shown in Figure 6 in the original paper.
> | Model            |     Accuracy     |
> | ---------------- |:----------------:|
> | Kaiming  |  86.30 $\pm$ 5.63   |
> | zero-padding     | 23.64 $\pm$ 0.55 |
> | identity-padding | 74.94 $\pm$ 0.91 |
>
> As mentioned by [2], Kaiming does not have a replication problem. Therefore, we use the forward and backward correlation of Kaiming as an indicator of trainability for a model. As shown in Table 2, the forward correlation of a zero-padding matrix is 0.00, indicating that zero-padding matrix based Allconv is not trainable. As a result, the zero-padding matrix can only derive 23.64% accuracy. By contrast, the identity-padding matrix solves the replication problem and derives a similar correlation in both forward and backward as the Kaiming initialization. Thus, the identity-padding matrix can derive better results.
>
> Although the identity-padding matrix is similar to the zero-padding matrix, the identity-padding matrix can solve both rank constraint and replication problems in the meantime. Therefore, we propose that the identity-padding matrix is an important contribution. As a result, IDI matrices (i.e., Eq. (7) in this paper) have sufficient differences from ZerO matrices (i.e., Definition 1 in [1]).

---

> > ### Author Response · Authors · 2022-11-19
> > **Response to reviewer Wd9d (Part 2)**
> >
> > **[R1S3]** Motivation on IDIC. The regular way to apply a matrix to convolution is formulated as Eq. (14) in our paper, which is also adopted in ZerO and ISONet. However, although the identity-padding matrix can solve rank constraint and resolve the replica problem [2], convolution from the identity-padding matrix by Eq. (14) still performs worse than the baseline Kaiming as shown in Table 3 and Table 4. Inspired by [3] increasing feature diversity among channels, and observing that Hadamard in ZerO can increase feature diversity too, we simply reshape $C$ in Eq. (15) to a convolution kernel, which is termed IDIC. We also apply a comparison between w/ IDIC and w/ IDIC on the AllConv experiment (settings are the same as that in Table 3 in the response) as follows.
> >
> > Table 4. Comparison between w/o IDIC and w/ IDIC. The big variance of Kaiming is caused by training instability as also shown in Figure 6 in the original paper.
> > | Model    | Accuracy |
> > | -------- |:--------:|
> > | Kaiming  |  86.30 $\pm$ 5.63 |
> > | w/o IDIC |  74.94 $\pm$ 0.91 |
> > | w/ IDIC  |  92.58 $\pm$ 0.20 |
> >
> > Apparently in above Table 4, IDIC can achieve significantly higher performance, which indicates enough effectiveness of IDIC. And this gap between w/ IDIC and w/o IDIC is called "Degeneration from Convolution Identity Transition" in Sec. 3.4, which is different from the "Training Degeneration" brought by rank constraint or replica problem as mentioned by prior studies[1]. In summary, this reshaping strategy of IDIC is completely different from the common convolution constructed as Eq. (14), and can make a large improvement on Eq. (14). Therefore, IDIC is supposed to be a valuable modification in our paper, which increases our novelty and contribution.
> >
> > **[R1S4]** Motivation on IDIZ. First of all, setting 0 as a multiplier in BN is implemented in [4]. Additionally, setting 0 or 1 for BN is the freedom of practitioners, and the goal of IDInit is to support any choice of practitioners. Therefore, we believe the 0-BN is a reasonable situation as one of the most famous deep model packages "timm" also sets the multiplier of the last BN to 0 by default. In addition, downsampling operations in ResNets can also generate features of full (https://github.com/akamaster/pytorch_resnet_cifar10/edit/master/resnet.py) or half 0 values (https://github.com/hongyi-zhang/Fixup/blob/master/cifar/models/resnet_cifar.py). All of these can lead to dead neural problems as we introduced in Sec. 3.3. Therefore, we propose IDIZ to initialize the last layer in a residual block to overcome this dead neural problem.
> >
> > We hope the above elaboration can resolve the concerns about the effectiveness of IDInit, and the motivation for IDIC and IDIZ.

---

> > > ### Author Response · Authors · 2022-11-19
> > > **Response to reviewer Wd9d (Part 3)**
> > >
> > > **[R2] Other Concerns**
> > > > (1) In addition, the authors only perform trainability analysis from a signal propagation perspective, without considering the potential degradation caused by the replication.
> > > >
> > > >(2) I highly recommend the authors to empirically evaluate if IDInit can pass the simple rank verification in Figure 3 of ZerO paper. On the other hand, I suspect the surprisingly good empirical evaluation of IDInit comes from randomized operations (e.g., dropout) in networks, which directly breaks the training degeneracy. I suggest the authors to carefully consider this training degeneracy issue, as it affects the effectiveness and universality of the proposed method.
> > >
> > > We have described the replication and rank constraint problems in [R1S2]. Removing dropout and "deterministic" setting, results show that identity-padding can solve both the replication and rank constraint problems as shown in Table 1 and Table 2 in the response.
> > >
> > > >The need of zero preserving initialization (IDIZ) is unclear.
> > >
> > > Please refer to [R1S4]. We have interpreted the motivation on IDIZ.
> > >
> > > >How does IDIC compare to the simple IDI in convolution case? The authors mention that IDIC is used to improve feature diversity but it's unclear why it is needed. ZerO paper [1] directly utilizes IDI in convolution (Algorithm 2 in ZerO) and it works well.
> > >
> > > Please refer to [R1S3]. We have interpreted the motivation on IDIC. ZerO does not only use zero-padding matrices. ZerO uses the Hadamard matrix to transform the zero-padding matrix to solve the rank constraint problem. However, this Hadamard transformation is not an identity transition operation, and also increases the feature diversity as IDIC does. Therefore, ZerO can work well with  Eq. (14).
> > >
> > > >Comparison to ZerO. Since ZerO is the most relevant work to IDInit I believe, I suggest the authors to empirically compare them, including both performance and determinacy analysis (since both methods claim the benefit as being deterministic).
> > >
> > > **Comparison of the Structure**
> > > 1. ZerO encounters the rank constraint since their zero-padding matrix will generate new zero features in transition. However, IDInit uses an identity-padding matrix. Although the identity-padding matrix seems slight difference from the zero-padding matrix, it can naturally avoid the harmful situation named rank constraint that causes training degeneracy in ZerO. We show a result on the same 3-layer experiment in ZerO as in Table 1, and derive a larger rank than 768, which indicates IDInit is not affected by the rank constraint problem. Therefore, this simple difference should be treated as an important contribution.
> > >
> > > 2. ZerO relies on the Hadamard matrix to relieve the rank constraint problem, which changes the identity transition in convolution. By contrast, we utilize the IDIC to reshape an identity-padding matrix. The reshaping operation will not change the identity transition, and increase feature diversity by shifting channel features to improve performance.
> > >
> > > 3. ZerO also simply set the last convolution in a residual stem to 0. However, this setting can cause a dead neural problem. They perform normally on ResNet-18/50 for the downsampling operation is a $1\times1$ convolution. However, in ResNet-20, 32, 56, 110, and so on, the downsampling operation always transits 0 in the residual connection, causing the dead neural problem. In addition, ZerO cannot be applied to a BN of 0 multiplier situation. These are all solved by our IDIZ initialization. Therefore, our method is more general.

---

> > > > ### Author Response · Authors · 2022-11-19
> > > > **Response to reviewer Wd9d (Part 4)**
> > > >
> > > > **Comparison of the Performance**
> > > >
> > > > First of all, we would to note that as deterministic initialization, both ZerO and IDInit will not keep the lower variance in experiments compared with random initialization. This reason is that the training framework is random for random batch sequence, random data crop, random data flip, and so on. Therefore, we will not compare the determinacy, but focus on the performance.
> > > >
> > > > Table 5. Intializaing ResNet-18 on Cifar-10.
> > > > | Model  | Accuracy         |
> > > > | ------ | ---------------- |
> > > > | ZerO   | 94.81 $\pm$ 0.05 |
> > > > | IDInit | 95.08 $\pm$ 0.13 |
> > > >
> > > > Table 6. Intializaing ResNet-56 on Cifar-10. ZerO performs worse for zero downsampling as mentioned in response of Reviewer st9W.
> > > > | Model  | SGD              | Adam             |
> > > > | ------ | ---------------- | ---------------- |
> > > > | ZerO   | 90.57 $\pm$ 0.31 | 83.53 $\pm$ 0.42 |
> > > > | IDInit | 93.41 $\pm$ 0.10 | 90.01 $\pm$ 0.32 |
> > > >
> > > > Table 7. Intializaing ResNet-110 on Cifar-10.
> > > > | Model  | SGD              | Adam             |
> > > > | ------ | ---------------- | ---------------- |
> > > > | ZerO   | 91.71 $\pm$ 0.21 | 84.24 $\pm$ 0.10 |
> > > > | IDInit | 94.04 $\pm$ 0.24 | 90.53 $\pm$ 0.10 |
> > > >
> > > > ResNet-18 (implementation URL: https://github.com/pytorch/vision/blob/main/torchvision/models/resnet.py) uses $1\times1$ convolution for downsampling. Therefore, there is no dead neuron problem in ResNet-18, and ZerO performs normally. Notably, IDInit can perform better than ZerO on ResNet-18. Compared to ResNet-18, ResNet-56 and ResNet-110 (implementation url: https://github.com/hongyi-zhang/Fixup/blob/master/cifar/models/resnet_cifar.py) increase the channel numbers by adding 0 values. Therefore, ZerO can encounter the dead neuron problem as mentioned in Sec. 3.3, and only derive 90.57% and 91.71% accuracies on ResNet-56 and ResNet-110, respectively. However, IDInit can solve this neuron problem and achieve much better results at 93.41% and 94.04%. Through these experiments, IDInit shows better performance and better flexibility than ZerO.
> > > >
> > > >
> > > > >However, I believe it lacks enough novelty. Identical initialization has been studied in ZerO paper [1], where IDInit and the proposed variant (Eq 14) in convolution are already considered. Its deterministic benefit is also studied in ZerO.
> > > >
> > > > As elaborated in [R1], motivation for IDIC and IDIZ is reasonable, and both two modifications can make a significant improvement on the basis identity-padding matrix. Therefore, our structure is clearly different from ZerO, and IDInit is not supposed to be a special case of ZerO. In addition, Eq. (14) is not our method thereby not affecting our novelty. Moreover, our experiments show better results and more flexible properties, which demonstrates our valid contribution and cannot be regarded as a part of ZerO. For determinacy, ZerO is only validated on ResNet and claims that determinacy can derive a lower performance variance. However, our paper finds this does not always hold as the random training framework affects performance a lot. In this paper, we construct experiments by fixing random factors like data batch sequence in Sec. 4.4. Therefore, we precisely observe the effectiveness of the determinacy. And we also validate determinacy on more structures, i.e., TextCNN, TextRNN and Transformer, on a TEXT CLASSIFICATION, not the image classification in ZerO. Therefore, we argue that our validation is more precise and broader. Considering all the above, we believe the proposed IDInit has sufficient novelty.
> > > >
> > > > [3] Kai, Han, et al. "Ghostnet: More features from cheap operations." Proceedings of the IEEE/CVF conference on computer vision and pattern recognition. 2020.
> > > >
> > > > [4] Priya, Goyal, et al. "Accurate, large minibatch sgd: Training imagenet in 1 hour." arXiv preprint arXiv:1706.02677 (2017).

---

### Decision · Program_Chairs · 2023-01-20

**Decision:**

Reject

**Justification For Why Not Higher Score:**

All reviewers agree this is a clear rejection

**Justification For Why Not Lower Score:**

N/A

**Metareview: Summary, Strengths And Weaknesses:**

The paper proposes a new initialization scheme called "identical initialization". Unfortunately, the reviewers clearly agree that the current form of the paper is below the bar for acceptance. Common concerns include poor quality of writing, unclear main message, unclear motivation for the proposed approach over FIXUP and ZerO (despite the authors trying to address this point). Some reviewers also point out some missing references.

I'm therefore not able to recommend acceptance, and I encourage the authors to address the concerns of the reviewers in a revised version.


**Summary Of Ac-Reviewer Meeting:**

N/A